# Can We Predict Performance of Large Models across Vision-Language Tasks?

Qinyu Zhao[1]  Ming Xu[1]  Kartik Gupta[2]  Akshay Asthana[2]  Liang Zheng[1]  Stephen Gould[1]

## Abstract

Evaluating large vision-language models (LVLMs) is very expensive, due to high computational cost and the wide variety of tasks. The good news is that if we already have some observed performance scores, we may be able to infer unknown ones. In this study, we propose a new framework for predicting unknown performance scores based on observed ones from other LVLMs or tasks. We first formulate the performance prediction as a matrix completion task. Specifically, we construct a sparse performance matrix $R$, where each entry $R_{mn}$ represents the performance score of the $m$-th model on the $n$-th dataset. By applying probabilistic matrix factorization (PMF) with Markov chain Monte Carlo (MCMC), we can complete the performance matrix, i.e., predict unknown scores. Additionally, we estimate the uncertainty of performance prediction based on MCMC. Practitioners can evaluate their models on untested tasks with higher uncertainty first, which quickly reduces the prediction errors. We further introduce several improvements to enhance PMF for scenarios with sparse observed performance scores. Our experiments demonstrate the accuracy of PMF in predicting unknown scores, the reliability of uncertainty estimates in ordering evaluations, and the effectiveness of our enhancements for handling sparse data. Our code is available at https://github.com/Qinyu-Allen-Zhao/CrossPred-LVLM.

## 1. Introduction

Large vision-language models (LVLMs) have shown remarkable performance across a wide range of multimodal

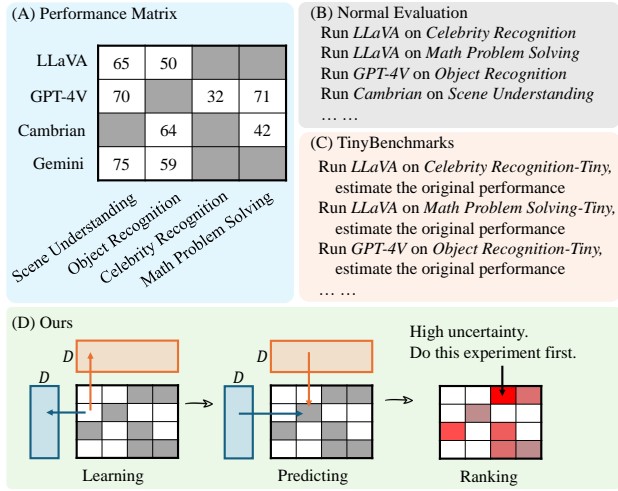

*Figure 1.* **Framework**. (A) Given a sparse matrix of performance scores of LVLMs on various tasks, the goal is to estimate the missing entries. (B) A normal way is to evaluate untested model-dataset pairs one-by-one. (C) TinyBenchmarks (Polo et al., 2024) runs models on smaller test sets and reproduce the original performance. (D) We use PMF to predict missing entries, reducing unnecessary evaluations, and rank new experiments based on uncertainty.

tasks (Liu et al., 2024; 2023a; Wang et al., 2024b; Bai et al., 2025; Lu et al., 2024; Chen et al., 2024; Achiam et al., 2023; Reid et al., 2024). However, due to their large scale and versatility, evaluating these models leads to high cost.

First, large-scale models result in significant computational and memory costs. Second, since a single LVLM can handle various tasks, hundreds of benchmarks have been proposed to comprehensively assess the strengths and weaknesses of models (Li & Lu, 2024; Li et al., 2023a; Fu et al., 2023). Last, increase in model size, the development of video LVLMs (Bai et al., 2025), and test-time scaling (OpenAI, 2024) techniques further raise evaluation cost. In our estimation, evaluating a 7B model on 50 benchmarks takes around 20 hours on one A100 GPU, and even three times longer when using test-time scaling or on video benchmarks (Fu et al., 2024; Wu et al., 2024) (Details in Section A).

Fortunately, we have already observed performance scores from some of these models on some tasks, for instance, from

[1]Australian National University, Canberra, Australia [2]Seeing Machines Ltd, Canberra, Australia. Correspondence to: Qinyu Zhao <qinyu.zhao@anu.edu.au>.

*Proceedings of the 42nd International Conference on Machine Learning*, Vancouver, Canada. PMLR 267, 2025. Copyright 2025 by the author(s).

the official reports of released models and datasets. For new models, scores can also be readily obtained with limited compute by running on a small number of tasks. If these observed scores can be used to predict unknown ones, we could avoid unnecessary evaluations and effectively reduce costs. Recent works (Polo et al., 2024; Zhang et al., 2024b) require running the same model on the same task to predict model performance, and most of them ignore the potential of observed performance data from other models or tasks.

In this study, we propose a new framework for predicting unknown performance scores based on observed ones from other LVLMs or tasks. We first formulate this as a matrix completion problem. Specifically, we construct a sparse performance matrix $R$ where each entry $R_{mn}$ represents the performance score of the $m$-th model on the $n$-th dataset. By applying probabilistic matrix factorization (PMF) with Markov chain Monte Carlo (MCMC), we can predict unknown performance scores based on observed entries in the matrix. A summary of the framework is shown in Fig. 1.

A bonus of our framework is active evaluation, which aims to select a subset of model-dataset pairs to evaluate in order to minimize prediction errors across the entire performance matrix. Given a PMF model on a very sparse performance matrix, we calculate prediction uncertainty from MCMC and prioritize evaluating model-dataset pairs with high uncertainty. Our experiments will confirm the effectiveness of this strategy for active evaluation.

A challenge is that PMF tends to predict the average score for models and datasets with very few observed scores, resulting in poor prediction results (Mnih & Salakhutdinov, 2007). To address this, we introduce several improvements to enhance PMF for scenarios with sparse observed data. First, we extend PMF to a simple tensor factorization approach, which can handle multiple performance metrics across different vision-language tasks. Second, we utilize Bayesian PMF (Salakhutdinov & Mnih, 2008) with an LKJ prior (Lewandowski et al., 2009) on the variance. Third, we also incorporate specific information from LVLMs and datasets to improve performance prediction. For example, if we know a model uses CLIP as a vision encoder, the information may help predict the model's performance, especially when we observe only a few performance scores.

In experiments, we conduct a systematic evaluation of 108 LVLMs across 176 distinct datasets derived from 36 existing benchmarks, based on prior works (Duan et al., 2024; Zhang et al., 2024b; Liang et al., 2024; Karamcheti et al., 2024; Schaeffer et al., 2024). We evaluate open-source models such as LLaVA-v1.5 (Liu et al., 2023a), Instruct-BLIP (Dai et al., 2023), mPLUG-Owl (Ye et al., 2023), and MiniGPT-4 (Zhu et al., 2023), as well as closed-source models including GPT-4o, GPT-4 (Achiam et al., 2023), Gemini-1.5 (Reid et al., 2024). The benchmarks cover gen-

eral VQA (Li et al., 2023a), knowledge-dense VQA (Yue et al., 2024), hallucination (Li et al., 2023b), medicine (He et al., 2020), emotion recognition (Goodfellow et al., 2013), and others. To reduce computational and API costs, we subsample some datasets, following Liang et al. (2024).

Using the results from 108 LVLMs across 176 datasets, we construct a $108 \times 176$ performance matrix, with some entries masked for testing. First, experiments demonstrate that PMF accurately predicts masked scores and consistently outperforms baselines as long as more than 10% entries in the performance matrix are observed. Second, we also show that selecting high-uncertainty model-dataset pairs for evaluation significantly reduces prediction errors compared to random selection. third, our improvements effectively alleviate the sparse data issue of PMF. Last, we also expand the performance matrix by introducing new models and datasets, showing the generalization ability of our method.

In summary, this paper covers three main points. First, we formulate a problem of predicting the unknown performance of LVLMs across tasks. Second, we apply the well-established PMF algorithm to this problem, show the application of active evaluation, and propose several strategies to mitigate the sparse data issue. Third, we conduct a comprehensive evaluation of LVLMs, constructing training and testing sets for further experiments.

## 2. Related Works

### 2.1. Benchmarking LVLMs

In recent years, an increasing number of LVLMs demonstrate impressive capabilities on a wide variety of tasks, including GPT-4 (Achiam et al., 2023), Gemini (Team et al., 2023), LLaVA (Liu et al., 2024; 2023a), Instruct-BLIP (Dai et al., 2023), InternVL (Chen et al., 2023; 2024), and Prismatic VLMs (Karamcheti et al., 2024). Their versatility requires various benchmarks to fully understand their strengths and weaknesses. Existing datasets can be repurposed for assessing these models, such as Flickr30k (Young et al., 2014), GQA (Hudson & Manning, 2019), and OKVQA (Marino et al., 2019). Recent works also propose new benchmarks to evaluate LVLMs in handling complex reasoning tasks, including SEED-Bench-2 (Li et al., 2023a), MMMU (Yue et al., 2024), and MME (Fu et al., 2023). Additionally, as LVLMs become more integrated into everyday applications, benchmarks (Li et al., 2023b) have been introduced to assess trustworthy issues like hallucination in these models. The variety of LVLMs and benchmarks leads to substantial computational demands and memory usage.

### 2.2. Improve Evaluation Efficiency

Recent works introduce unified frameworks to assess models across multiple benchmarks using a single code-

base, such as VLMEvalKit (Duan et al., 2024), LMMs-Eval (Zhang et al., 2024b), and HEMM (Liang et al., 2024). Our study builds on these efforts by consolidating their frameworks and integrating models in Prismatic VLMs series (Karamcheti et al., 2024; Schaeffer et al., 2024).

Predicting unknown model performance can reduce the evaluation cost. Prior works select a coreset of samples from a large benchmark, for evaluating large language models (LLMs) (Polo et al., 2024; Perlitz et al., 2023) and LVLMs (Zhang et al., 2024b; Zhu et al., 2024). The performance of a specific model on the coreset is used to estimate its performance on the full benchmark. Besides, existing studies estimate model performance on an unlabeled test set based on distribution shift (Deng & Zheng, 2021), confidence scores (Guillory et al., 2021; Yang et al., 2024), or LLM feedback (Zheng et al., 2023). Instead of running models on a coreset or an unlabeled set, our framework predicts unknown performance by utilizing the correlation between model performances across benchmarks, and is complementary to coreset-based ones (see Section 5.3).

A concurrent study by Zhang et al. (2024c) applies deep matrix factorization to predict LLM performance. Their goal and methodology differ from ours. First, their work aims to explore whether LLM performance can be predicted except using scaling laws, while we focus on improving evaluation efficiency. Second, they use neural networks for prediction, whereas we adopt PMF with MCMC, which allows us to quantify uncertainty in performance prediction.

Another related direction is adaptive testing (Rodriguez et al., 2021; Prabhu et al., 2024), i.e., adaptively selecting a subset of samples for evaluation. While prior work relies on statistically inferred sample difficulty, we propose a method to rank model-dataset pairs for evaluation based on uncertainty in performance prediction from MCMC.

### 2.3. Probabilistic Matrix Factorization

PMF (Mnih & Salakhutdinov, 2007) is widely applied in recommender systems. Given part of user ratings for items, the goal is to model the observed ratings and predict the missing ones. PMF achieves this by decomposing the observed rating matrix into two lower-dimensional matrices, representing the latent features of users and items. A rating is modeled via a Gaussian distribution centered around the dot product of the user's and item's feature vectors.

One major challenge with PMF is that, if users rate very few items, their predicted ratings will be near the average for those items. Bayesian PMF (BPMF) (Salakhutdinov & Mnih, 2008) addresses this by placing distributions over the priors of the latent user and item features, making it more effective in handling sparse data. Additionally, Constrained PMF (Mnih & Salakhutdinov, 2007) introduces a latent

similarity constraint matrix to address the problem.

## 3. Modeling LVLM Performance

In this section, we first describe the application of PMF to model the performance score matrix of LVLMs across datasets. Then, we discuss active evaluation for LVLMs. Last, three techniques are introduced to enhance PMF: supporting multiple metrics, incorporating Bayesian PMF, and integrating model and dataset profiles in modeling.

### 3.1. Revisit PMF

Let $\boldsymbol{R}$ be an $M \times N$ matrix representing model performance scores on datasets, where $M$ is the number of models and $N$ is the number of datasets. For simplicity, we initially assume a single performance metric. In reality, benchmarks often employ multiple metrics, so $\boldsymbol{R}$ becomes an $M \times N \times S$ tensor, where $S$ represents the total number of metrics. We will extend our framework to this in Section 3.3.

In practice, only a subset of the elements in $\boldsymbol{R}$ are observed, meaning we evaluate only a portion of the model-dataset pairs and aim to estimate the remaining performance scores. Specifically, we define a matrix $\boldsymbol{O} \in \{0, 1\}^{M \times N}$, where $O_{mn} = 1$ if $R_{mn}$ is observed, and 0 otherwise.

To model the observed matrix and estimate the unknown values, we employ PMF (Mnih & Salakhutdinov, 2007), as illustrated by the probabilistic graphical model in Fig. 2(A). PMF decomposes $\boldsymbol{R}$ into two low-dimensional matrices, $\boldsymbol{U} \in \mathbb{R}^{M \times D}$ and $\boldsymbol{V} \in \mathbb{R}^{N \times D}$, where $D$ is the latent dimension. Here, $\boldsymbol{U}_{m,:}$ and $\boldsymbol{V}_{n,:}$ are the latent feature vectors for the $m$-th model and the $n$-th dataset, respectively, and we refer to them as $\boldsymbol{U}_m$ and $\boldsymbol{V}_n$. These latent vectors are modeled as multivariate Gaussian distributions, and the observed ratings are assumed to follow a Gaussian distribution centered at the dot product of the latent feature vectors:

$$p(\boldsymbol{R} \mid \boldsymbol{U}, \boldsymbol{V}, \sigma^2) =$$
$$\prod_{m=1}^{M} \prod_{n=1}^{N} \left[ \mathcal{N}\left( R_{mn} \mid \boldsymbol{U}_m^T \boldsymbol{V}_n, \sigma^2 \right) \right]^{O_{mn}}, \qquad (1)$$

$$p(\boldsymbol{U} \mid \sigma_U^2) = \prod_{m=1}^{M} \mathcal{N}(\boldsymbol{U}_m \mid \boldsymbol{0}_D, \sigma_U^2 \boldsymbol{I}_D), \qquad (2)$$

$$p(\boldsymbol{V} \mid \sigma_V^2) = \prod_{n=1}^{N} \mathcal{N}(\boldsymbol{V}_n \mid \boldsymbol{0}_D, \sigma_V^2 \boldsymbol{I}_D), \qquad (3)$$

where $\boldsymbol{I}_D$ is a $D \times D$ identity matrix, and $\mathcal{N}(x \mid \mu, \sigma^2)$ represents the probability density function of a Gaussian distribution with mean $\mu$ and variance $\sigma^2$. For simplicity, we set $\sigma_U = \sigma_V = 1$.

Rather than using Maximum A Posteriori estimation to obtain point estimates of the unknown performance scores in

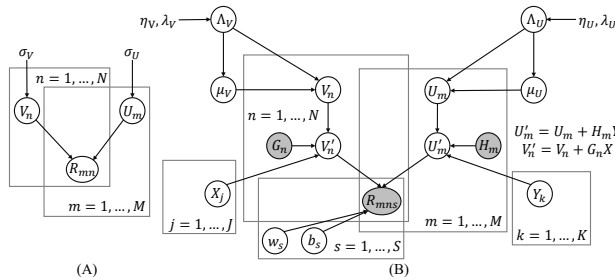

(A)  (B)

*Figure 2.* **Graphical Models** of PMF (A) and the enhanced model (B). (A) is adapted from the original paper (Mnih & Salakhutdinov, 2007). In (B), we set the mean to **0** and the covariance to the identity matrix, thus omitting most of the hyper-parameters for the random variable distributions.

$R$, we apply MCMC to obtain distributions over the estimated scores and quantify prediction uncertainty.

We compare PMF and deep learning-based matrix completion methods, such as Deep Matrix Factorization (DMF) (Arora et al., 2019) and Graph Convolutional Matrix Completion (GCMC) (Berg et al., 2017). The results indicate that PMF is more suitable to our problem than deep learning methods, probably because deep learning is more data hungry. Thus, this study will focus on PMF. The details are shown in the supplementary materials (Section C.1).

### 3.2. Active Evaluation

MCMC allows us to estimate score distributions and readily obtain uncertainty estimates for each unknown score, enabling us to prioritize evaluation experiments. For example, if we are uncertain about GPT-4's performance on 3D understanding but confident about LLaVA's performance on object recognition, we can prioritize evaluating GPT-4 on the 3D task when our resources are limited.

In our method, we begin by applying PMF to model a sparse performance matrix. Using MCMC, we get hundreds of estimations of each unknown score and calculate the standard deviation of estimations as a measure of uncertainty. The unobserved scores are ranked by their uncertainties. High-uncertainty scores are replaced with ground truth, simulating evaluation process in practice. We rerun PMF with updated observed data, calculate uncertainty, and determine the next set of evaluations. This process is repeated until our resource budget is exhausted or all scores are observed.

### 3.3. Multiple Metrics

Our experiments show that standard PMF performs well with sufficient observed data. But its performance degrades significantly and is even worse than predicting the average score, when the observed data is very sparse (i.e., fewer

than 10% model-dataset pairs are observed). To address this, we enhance our model with several techniques, with a new graphical model shown in Fig. 2(B).

Previously, we assumed that each dataset has only one scoring metric, but this is usually not the case in practice. For example, yes-or-no questions can be evaluated using accuracy, precision, recall, and F1 score, while open-ended questions may use metrics like BART score (Yuan et al., 2021) and BERT score (Zhang et al., 2019). Model performances are represented by a tensor $\mathbf{R} \in \mathbb{R}^{M \times N \times S}$, where $S$ is the total number of metrics. Empirically, we find that using PMF to model and predict each metric independently works well when sufficient data is available. However, when observed data is sparse, incorporating relationships between metrics will be helpful to performance prediction.

To address this, we extend our PMF model into a simple Probabilistic Tensor Factorization (PTF), where we decompose the 3D tensor $\mathbf{R}$ into the product of two low-rank matrices and a 1D vector. This can be interpreted as applying a linear transformation to the original PMF output, translating it into multiple metrics. Specifically, we define:

$$p(\mathbf{R} \mid \boldsymbol{U}, \boldsymbol{V}, \boldsymbol{w}, \boldsymbol{b}, \sigma^2) =$$
$$\prod_{m=1}^{M} \prod_{n=1}^{N} \prod_{s=1}^{S} \left[ \mathcal{N}\left( R_{mns} \mid (\boldsymbol{U}_m^T \boldsymbol{V}_n) w_s + b_s, \sigma^2 \right) \right]^{O_{mns}},$$
$$(4)$$

$$p(\boldsymbol{w} \mid \sigma_w^2) = \mathcal{N}(\boldsymbol{w} \mid \mathbf{0}_S, \sigma_w^2 \boldsymbol{I}_S), \qquad (5)$$
$$p(\boldsymbol{b} \mid \sigma_b^2) = \mathcal{N}(\boldsymbol{b} \mid \mathbf{0}_S, \sigma_b^2 \boldsymbol{I}_S), \qquad (6)$$

where we set $\sigma_w = \sigma_b = 1$ for simplicity.

This approach implicitly assumes a linear relationship between scoring metrics, which may not exactly hold in reality. However, we usually observe some linear correlation between the metrics on the same task. More sophisticated techniques, such as advanced tensor factorization methods are left as future work to further improve the model.

Note that some metrics may be irrelevant for certain datasets, e.g., accuracy is not meaningful for long-answer questions. While our model can predict these scores, we discard the predicted results because there is no ground truth.

### 3.4. Bayesian PMF

Instead of using fixed priors for the feature vectors, we model the priors using probabilistic distributions, as proposed by Salakhutdinov & Mnih (2008). Unlike the original paper, which employs a Wishart distribution for the variance, we use the LKJ correlation prior (Lewandowski et al., 2009) and an Exponential prior to model the variance, as suggested by the PyMC official documentation,

$$\Lambda_U^{-1} = (\mathbf{diag}\,(\boldsymbol{\sigma}_L)\,\boldsymbol{L}_U)(\mathbf{diag}\,(\boldsymbol{\sigma}_L)\,\boldsymbol{L}_U)^T, \qquad (7)$$

where $p(\boldsymbol{L}_U \mid \eta_U) = \mathrm{LKJ}(\boldsymbol{L}_U \boldsymbol{L}_U^T \mid \eta_U)$ and $p(\boldsymbol{\sigma}_L \mid \lambda_U) = \prod_{d=1}^{D} \mathrm{Exp}(\sigma_d \mid \lambda_U)$. Then, latent feature vectors are modeled as follows:

$$p(\boldsymbol{\mu}_U \mid \boldsymbol{\Lambda}_U^{-1}) = \mathcal{N}(\boldsymbol{\mu}_U \mid \boldsymbol{0}_D, \boldsymbol{\Lambda}_U^{-1}), \tag{8}$$

$$p(\boldsymbol{U} \mid \boldsymbol{\mu}_U, \boldsymbol{\Lambda}_U^{-1}) = \prod_{m=1}^{M} \mathcal{N}(\boldsymbol{U}_m \mid \boldsymbol{\mu}_U, \boldsymbol{\Lambda}_U^{-1}). \tag{9}$$

A similar formulation applies to $\boldsymbol{V}$, which we omit here.

### 3.5. Model and Dataset Profiles

The final enhancement is the incorporation of additional information about the models and datasets. For example, knowing that two LVLMs use CLIP as the vision encoder, or that LLaVA-v1.5 and LLaVA-NeXT are developed by the same team, suggests potential relationships in their performances. Inspired by Constrained PMF (Mnih & Salakhutdinov, 2007), we incorporate extra information as model and dataset profiles, to improve performance prediction.

Let $\boldsymbol{H} \in \mathbb{R}^{M \times K}$ and $\boldsymbol{G} \in \mathbb{R}^{N \times J}$ represent the model and dataset profiles, where $\boldsymbol{H}_{m,:}$ encodes $K$ properties of the $m$-th model (e.g., vision encoder type), and $\boldsymbol{G}_{n,:}$ encodes $J$ properties of the $n$-th dataset. We introduce Gaussian-distributed variables $\boldsymbol{Y} \in \mathbb{R}^{K \times D}$ and $\boldsymbol{X} \in \mathbb{R}^{J \times D}$ to learn the effects of these profiles. The latent feature vectors are now the sum of the original vectors and the profile features, following Constrained PMF (Mnih & Salakhutdinov, 2007).

$$p(\boldsymbol{Y} \mid \sigma_Y^2) = \prod_{k=1}^{K} \mathcal{N}(\boldsymbol{Y}_k \mid \boldsymbol{0}_D, \sigma_Y^2 \boldsymbol{I}_D), \tag{10}$$

$$p(\boldsymbol{X} \mid \sigma_X^2) = \prod_{j=1}^{J} \mathcal{N}(\boldsymbol{X}_j \mid \boldsymbol{0}_D, \sigma_X^2 \boldsymbol{I}_D), \tag{11}$$

$$\boldsymbol{U}' = \boldsymbol{U} + \boldsymbol{H}\boldsymbol{Y}, \quad \boldsymbol{V}' = \boldsymbol{V} + \boldsymbol{G}\boldsymbol{X}. \tag{12}$$

**Oracle Profiles.** To explore the upper bound of model and dataset similarities, we use the full $\boldsymbol{R}$ matrix to cluster models and datasets. For each model, we take $\boldsymbol{R}_{i,:}$ (its performance across all datasets) as a vector and apply the K-Means algorithm to cluster all models. We select the optimal number of clusters using the elbow method. Similarly, for each dataset, we cluster $\boldsymbol{R}_{:,j}$ in the same way. We convert the cluster assignments into one-hot vectors as profiles.

**Custom Profiles.** Since oracle profiles rely on complete performance data, they are not feasible for real-world use. To overcome this, we define custom profiles that can be applied in practice. For models, we include features such as the number of parameters in the LLM backbone, vision encoder type (one-hot), and the LVLM family (one-hot), illustrated in the supplementary material (Table 8). Additionally, we cluster datasets based on latent representations obtained from various models and get one-hot encoded dataset profiles. We

explore three different approaches to generate these latent representations: D1. using MPNet (Song et al., 2020) to encode a short description of each dataset. D2. using CLIP to encode images and BGE-M3 to encode questions in a dataset (following Zhang et al. (2024b)), then averaging the embeddings on the dataset; and D3. using LLaVA-7B to encode both images and text, then averaging the embeddings as the profile for the dataset.

## 4. Experiments

We first construct a large performance matrix by evaluating LVLMs on different benchmarks, and then, we present key experiments to validate our framework.

### 4.1. Evaluating Models on Benchmarks

Based on prior code repositories (Duan et al., 2024; Zhang et al., 2024b; Liang et al., 2024), we evaluate 108 LVLMs on 176 tasks in 36 benchmarks and build a $108 \times 176$ performance matrix, which is used for our main experiments on PMF and MCMC in Section 4.2 and 4.3.

The open-source models we cover include LLaVA-v1.5 (Liu et al., 2023a), LLaVA-NeXT (Liu et al., 2023a), Instruct-BLIP (Dai et al., 2023), mPLUG-Owl (Ye et al., 2023), and Prismatic VLMs (Karamcheti et al., 2024). We also evaluate closed-source models like GPT-4 (Achiam et al., 2023) and Gemini-1.5 (Reid et al., 2024).

The benchmarks span a variety of domains, including general VQA (SEED-2), knowledge-dense VQA (MMMU), hallucination (POPE), medical question answering (PathVQA), and emotion recognition (FaceEmotion). Some large-scale benchmarks, such as SEED-2 (Li et al., 2023a) and MMMU (Yue et al., 2024), cover multiple tasks. To conduct a fine-grain analysis, we split these benchmarks into task-specific datasets, resulting in 176 datasets in total. Following HEMM (Liang et al., 2024), we subsample some datasets to reduce computational and API calling costs of LVLMs. For each dataset, we calculate a main metric for PMF (either accuracy or BARTScore), and several other metrics, leading to a total of six metrics for PTF modeling.

To evaluate the generalization ability of our framework, we expand the model and dataset pool by introducing **new models**: Qwen2-VL-Instruct 2B / 7B (Wang et al., 2024b), Qwen2.5-VL-Instruct 3B / 7B / 32B (Bai et al., 2025), and DeepSeek-VL tiny / small (Lu et al., 2024), as well as **new benchmarks**: MathVision (Wang et al., 2024a), EMMA (Hao et al., 2025), Video-MME (Fu et al., 2024), and LongVideoBench (Wu et al., 2024). This expansion increases the performance matrix to cover 115 LVLMs across 40 benchmarks. We present the results in Section 4.5.

Full details of datasets and models are provided in the sup-

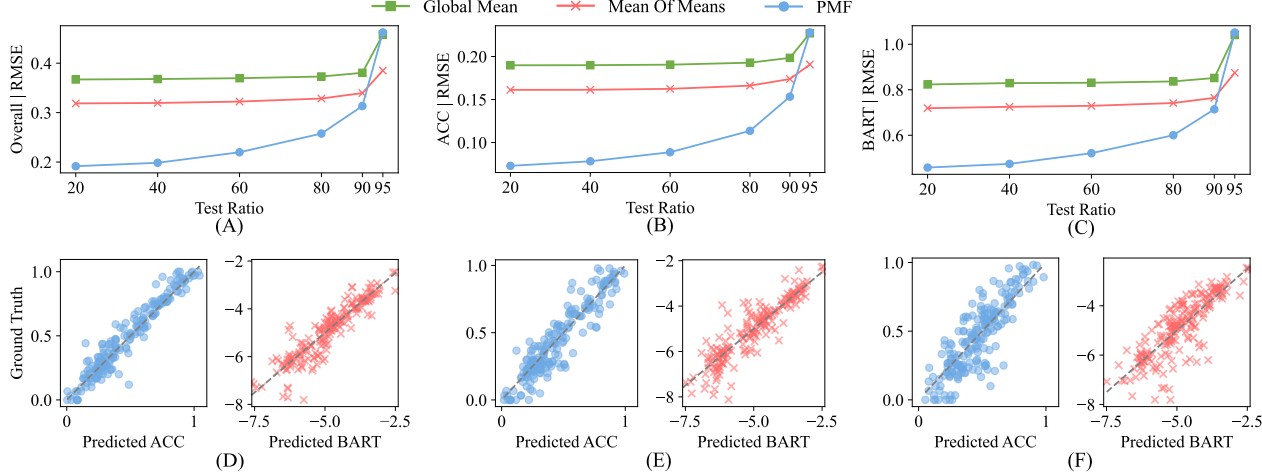

*Figure 3.* **Performance of PMF.** (A-C) PMF consistently outperforms both baselines when the test ratio is below 90% for estimating all unobserved scores (A), accuracy scores (B), and BART scores (C), with particularly strong performance at lower test ratios. (D-F) The predicted scores exhibit correlations with the ground truth at test ratios of 20% (D), 60% (E), and 90% (F). Gray dashed lines represent perfect prediction i.e., $y = x$. We subsampled 200 scores in (D-F) for visualization.

plementary material (Section B).

### 4.2. Estimating Unknown Performance

We mask $P\%$ of the elements in the score matrix $\boldsymbol{R}$, use the observed portion to normalize $\boldsymbol{R}$, and train the PMF model using MCMC sampling. The model reconstructs the matrix $\hat{\boldsymbol{R}}$, and we evaluate the performance by comparing the estimated values with the ground truth for the masked elements. For MCMC, we use the No-U-Turn Sampler (NUTS) (Hoffman et al., 2014), an advanced Hamiltonian Monte Carlo method (Neal, 2011), tuning with 500 samples in the burn-in stage and drawing 100 samples. Empirical results show that 100 samples are sufficient for stable estimation. The mean prediction from MCMC is the final reconstructed matrix $\hat{\boldsymbol{R}}$.

We use Root Mean Squared Error (RMSE) as the primary metric. Additional metrics such as Mean Absolute Error (MAE) and the coefficient of determination ($R^2$), as well as ranking-based metrics including Spearman's rank correlation, Kendall rank correlation, Precision and Recall are reported in the supplementary material (Section C).

We compare our method against two baselines: (1) Global Mean: predicting the global mean for unobserved scores; (2) Mean of Means: for each unobserved score, we average the mean performance of the model, the mean performance on the dataset, and the global mean.

**Results.** As shown in Fig. 3(A-C), PMF significantly outperforms the baselines when the test ratio is lower than 90%. This suggests that when only a portion of the scores is available, PMF can infer the unobserved scores with high accuracy. Additionally, as demonstrated in Fig. 3(D-F), the

*Table 1.* **Comparison of PMF and PTF.** Superior results are highlighted. PMF (Sep) models each score separately, while PMF (OneMat) combines accuracy and BART scores into a single matrix, as each dataset contains either accuracy or BART scores. PTF is the enhanced model that supports multiple scoring metrics, which outperforms PMF at a high test ratio.

| Method | Overall | Acc | Precision | Recall | F1 | BART | BERT |
|---|---|---|---|---|---|---|---|
| *Test Ratio: 20%* | | | | | | | |
| PMF (Sep) | 0.175 | 0.073 | 0.135 | 0.166 | 0.134 | 0.463 | 0.068 |
| PMF (OneMat) | 0.193 | 0.074 | - | - | - | 0.461 | - |
| PTF | 0.205 | 0.078 | 0.129 | 0.176 | 0.108 | 0.563 | 0.077 |
| *Test Ratio: 90%* | | | | | | | |
| PMF (Sep) | 0.327 | 0.159 | 0.238 | 0.262 | 0.227 | 0.864 | 0.096 |
| PMF (OneMat) | 0.317 | 0.156 | - | - | - | 0.723 | - |
| PTF | 0.290 | 0.159 | 0.186 | 0.230 | 0.180 | 0.754 | 0.094 |

estimated scores strongly correlate with the actual scores, indicating reliable predictions from our framework.

However, as the amount of observed data decreases, PMF's performance declines as can be expected. In extreme cases where the test ratio exceeds 90%, with limited information about model or dataset performance, PMF can perform worse than predicting the means.

### 4.3. Active Evaluation for LVLMs

We compare our uncertainty-based approach against two baselines: (1) random selection of model-dataset pairs, and (2) an oracle approach that selects the pairs with the highest actual errors. In the experiment, we start by 20% data for initial training, 60% as the pool set, and 20% for testing. Then, we progressively include more data from the pool set for training using three different strategies, and calculate

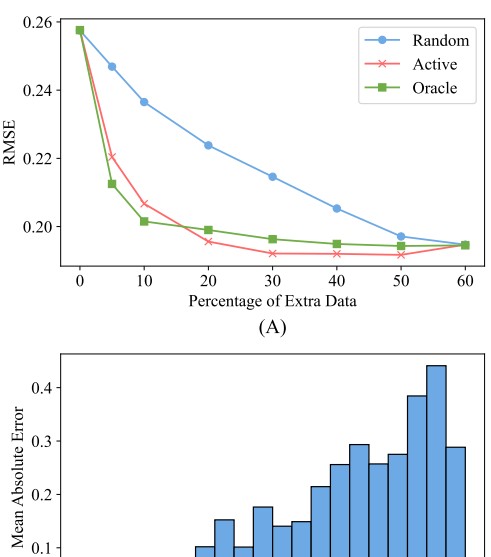

(A)

(B)

*Figure 4.* **Comparison of Active Evaluation Methods.** Starting with 20% of the data observed, we progressively conduct additional LVLM evaluations using three different strategies. (A) RMSE improvement demonstrate the advantage of our method compared to random evaluation. (B) Uncertainties from MCMC are correlated with the actual absolute errors.

the improvement in performance prediction on the test data. The experiment is repeated with 10 different random seeds, and we report the average improvement.

**Results.** As shown in Fig. 4, our uncertainty-aware method consistently outperforms the random baseline for a fixed budget of evaluations, especially when the amount of extra data is lower than 30%. Besides, uncertainties from MCMC are correlated with the actual absolute errors.

### 4.4. Enhancing PMF

We apply three enhancement techniques to our PMF model and evaluate their effectiveness across different test ratios. To minimize experimental variance, we perform each experiment 10 times with different random seeds and report the average performance at each test ratio.

**Results.** As seen in Table 1, the multi-score method PTF can get better performance when the matrix is very sparse. When there is enough data, separately modeling PMF with each score works very well and is comparable to PTF. For BART and BERT scores, PMF even outperforms PTF. This is likely because PTF assumes a linear relationship between scores. When this assumption does not hold, such as in the case of BART and BERT scores, it degrades performance.

*Table 2.* **Generalization to New Models and Datasets.** Average RMSEs of different methods are reported.

| Method | New Model | New Dataset | Both New |
|---|---|---|---|
| *If we only know 20% performance of new models and new datasets* | | | |
| Global Mean | 0.390 | 0.043 | 0.106 |
| Mean of Means | 0.303 | 0.037 | 0.084 |
| PMF | 0.326 | 0.032 | 0.047 |
| BCPMF | 0.297 | 0.033 | 0.039 |
| *If we know 50% performance of new models and new datasets* | | | |
| Global Mean | 0.389 | 0.045 | 0.090 |
| Mean of Means | 0.311 | 0.039 | 0.073 |
| PMF | 0.265 | 0.031 | 0.034 |
| BCPMF | 0.228 | 0.030 | 0.034 |

Fig. 5 illustrates the impact of the other two enhancement techniques. As shown, Bayesian PTF offers only negligible improvements over standard PTF when there is enough observed data, but it is particularly beneficial in sparse conditions. In Fig. 5(B), our custom profiles also show improvements when data is limited, though there remains a gap between our custom profiles and the oracle profiles. Additionally, Fig. 5(C) highlights that adding profiles not only enhances PTF's overall performance but also reduces instability, as seen by smaller error bars. Model profiles show significant performance gains, whereas dataset profiles contribute only marginally. Better methods for encoding and utilizing dataset information need further exploration.

### 4.5. Generalization to New Models and Datasets

To validate the generalization ability of the framework, we expand the performance matrix by adding new models and datasets. Then, we provide the complete performance data on old models and datasets, along with partial observations for: (1) old models on new datasets, (2) new models on old datasets, and (3) new models on new datasets. Based on this setup, we predict the missing performance scores and calculate the average RMSEs for different methods.

As shown in Table 2, our method shows better generalization compared to the baselines. We observe that generalizing to new datasets is relatively easier than to new models. This is probably because new datasets are often very challenging, leading to generally lower model performance and smaller RMSEs. In contrast, new models often have novel designs and remarkable improvements. This makes generalization to unseen models more difficult and needs further exploration.

## 5. Discussion

### 5.1. Predictability of Model Performance

There are two possible perspectives to understand why the model performance can be well predicted.

**Benchmarks.** There is redundancy in existing benchmarks, as also reported by Zhang et al. (2025). Multiple bench-

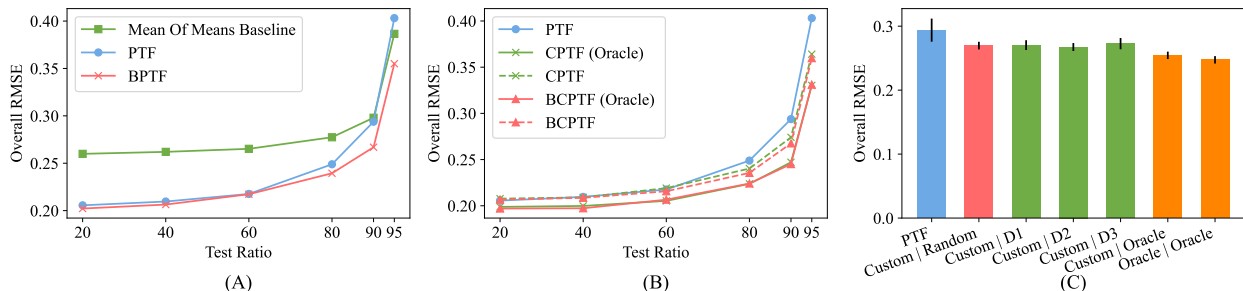

(A)           (B)           (C)

*Figure 5.* **Performance of Enhanced PTF.** (A) BPTF shows minimal improvement over standard PTF when data is sufficient but proves particularly beneficial under sparse conditions. (B) Custom profiles improve performance when data is limited, though a gap remains compared to oracle profiles. (C) Ablation study on model and dataset profiles. "A | B" represents using A for the model profile and B for the dataset profile. Custom model profiles lead to significant performance gains, while dataset profiles contribute only marginally. BPTF, Bayesian PTF; CPTF, Constrained PTF.

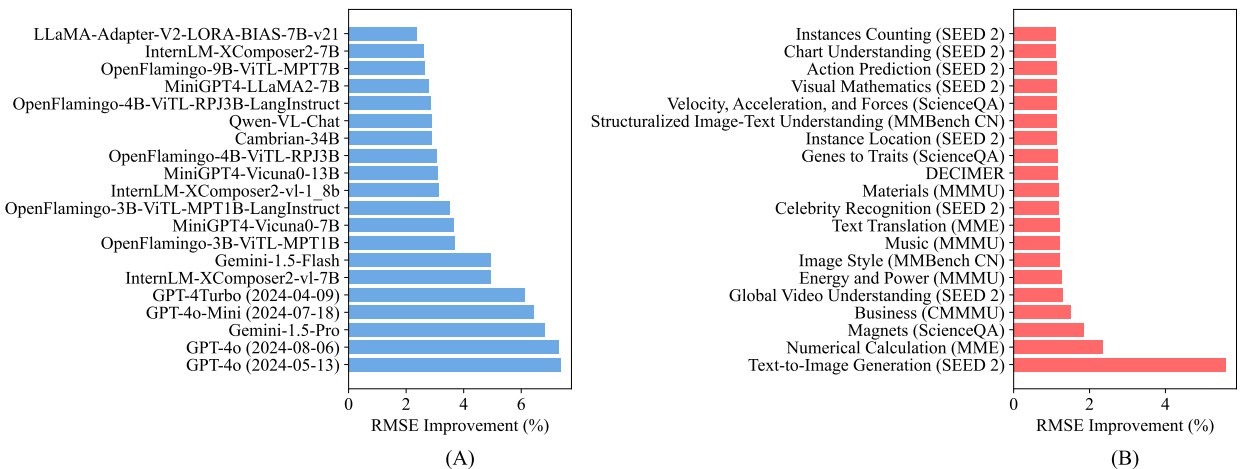

(A)                                  (B)

*Figure 6.* **Which Models and Datasets Are Informative for Performance Estimation.** Given a PMF model train on 20% data of the performance matrix, We measure the improvement in RMSE of PMF when adding the entire results of a model (A) or a dataset (B).

marks may test similar skills like math. Besides, there might be correlation across different tasks. For instance, Liu et al. (2023a) reports that an LVLM with a stronger LLM can achieve consistently better performance on different tasks.

**Models.** Similarity in model architectures and training may also contribute to correlation between performance. First, if models use the same vision encoder, e.g., CLIP, they may share similar failure cases (Tong et al., 2024b). Besides, many models use training data generated from GPT-4V, possibly resulting in similar strengths and weaknesses.

Predictability is valuable during model development. For example, when exploring optimal video LLM design, Zohar et al. (2024) evaluate their models on a reduced set of benchmarks to guide design decisions. Only at the final stage, they assess the model on the full benchmarks to report actual performance. In such cases, they could use our method to reduce unnecessary evaluation when developing models.

## 5.2. Modeling Varying Settings of LVLM Evaluation.

Our framework is also extensible to different evaluation settings, such as various prompts or decoding strategies. There are two possible methods. **Additional models:** treat a model under different evaluation settings as different models, such as "LLaVA (Chain-of-Thought)" and "LLaVA (Beam Search)". **Additional profile:** encode evaluation settings as extra profiles into PMF.

Specifically, we evaluate LLaVA-v1.5-7B on the 27 tasks in SEED-2, with the following various evaluation settings.

**Image input.** (1) Default: use the clean images, or (2) add Gaussian noise into the images.

**Prompt.** (1) Default: prompt the model to choose option ("Answer with the option's letter from the given choices directly."), (2) provide no hint, or (3) use the Chain-of-Thought (CoT) prompt ("Let's think step by step, and then

*Table 3.* **Prediction Accuracy under Different Evaluating Settings.** Average RMSEs of different methods are reported.

| Method | Default | Input | Prompt | | Decoding | | |
|---|---|---|---|---|---|---|---|
| | | Gaussian Noise | No Hint | CoT | Sampling (t=0.2) | Sampling (t=0.5) | Beam |
| *Test Ratio: 20%* | | | | | | | |
| Global Mean | 0.112 | 0.105 | 0.090 | 0.117 | 0.127 | 0.109 | 0.111 |
| Mean of Means | 0.090 | 0.088 | 0.090 | 0.102 | 0.105 | 0.092 | 0.088 |
| Ours (Profiles) | 0.041 | 0.055 | 0.075 | 0.064 | 0.045 | 0.055 | 0.052 |
| Ours (Models) | 0.043 | 0.045 | 0.073 | 0.050 | 0.040 | 0.046 | 0.041 |
| *Test Ratio: 80%* | | | | | | | |
| Global Mean | 0.140 | 0.115 | 0.093 | 0.115 | 0.131 | 0.132 | 0.139 |
| Mean of Means | 0.119 | 0.097 | 0.094 | 0.099 | 0.109 | 0.114 | 0.123 |
| Ours (Profiles) | 0.089 | 0.099 | 0.090 | 0.096 | 0.082 | 0.111 | 0.117 |
| Ours (Models) | 0.094 | 0.081 | 0.092 | 0.088 | 0.075 | 0.092 | 0.095 |

*Table 4.* **Performance of Combining Coreset Methods with Our Approach.** As seen, coreset methods and our approach are complementary. Their combination achieves better performance than either method individually.

| Method | Overall | | Acc | | BART | |
|---|---|---|---|---|---|---|
| | RMSE↓ | MAE↓ | RMSE | MAE | RMSE | MAE |
| Ours | 0.193 | 0.090 | 0.074 | 0.052 | 0.459 | 0.299 |
| Random Selection | 0.224 | 0.141 | 0.152 | 0.107 | 0.444 | 0.326 |
| + Ours (Avg) | 0.149 | 0.088 | 0.085 | 0.061 | 0.322 | 0.237 |
| + Ours (Unc) | 0.139 | 0.076 | 0.069 | 0.049 | 0.313 | 0.224 |
| Herding | 0.216 | 0.140 | 0.155 | 0.112 | 0.410 | 0.297 |
| + Ours (Avg) | 0.144 | 0.088 | 0.087 | 0.064 | 0.305 | 0.220 |
| + Ours (Unc) | 0.140 | 0.076 | 0.070 | 0.049 | 0.315 | 0.220 |
| K-Center Greedy | 0.223 | 0.142 | 0.154 | 0.109 | 0.437 | 0.322 |
| + Ours (Avg) | 0.144 | 0.088 | 0.086 | 0.063 | 0.306 | 0.226 |
| + Ours (Unc) | 0.138 | 0.077 | 0.070 | 0.049 | 0.313 | 0.224 |

answer with the option's letter.").

**Model decoding.** (1) Default: greedy decoding, (2) sampling with temperature = 0.2, (3) sampling with temperature = 0.5, or (4) beam search with temperature = 0.2 and the number of beams = 10.

We add the results under different evaluation settings into our framework and simply use PMF for prediction. Table 3 reports RMSE of different methods and indicate that our framework can handle different evaluation settings.

### 5.3. Combine Coreset Method with Our Approach

Previous studies (Polo et al., 2024; Zhang et al., 2024b) select a small set of representative samples as a coreset in a benchmark and evaluate models on the coreset. Our work utilizes known model performance from different benchmarks or models and is complementary to existing methods.

In experiments, we explore two combination approaches. (1) Avg: simply average predictions of coreset methods and PMF; (2) Unc: use uncertainties from MCMC to combine the coreset and PMF predictions. In short, when PMF is confident, we mainly rely on using known performance for prediction, and otherwise, rely on the coreset methods.

Following Zhang et al. (2024b), we use CLIP to generate

embeddings for images and BGE-M3 for text, and concatenates them to get the samples embeddings. Based on sample embeddings, we use random, Herding, and K-Center Greedy to select 10% core samples from benchmarks.

Table 4 demonstrates the effectiveness of our approach. More experimental results can be found in the supplementary materials (Section C.3).

### 5.4. What Can We Tell Based on Vision Encoders?

Constrained PMF can capture the impact of model and dataset profiles. Here, we present a showcase analysis focusing on the impact of vision encoder type. Specifically, we calculate the dot product between the feature vector of the vision encoder type, $H_m$, and the feature vector of the dataset, $V'_n$. The calculation result measures the influence of a vision encoder on a task. In experiments, CLIP and DINO show improvements on a few datasets, while FNet, SigLIP, and ViT are less effective in comparison. The details are shown in the supplementary material (Section C.4).

### 5.5. Which Models or Benchmarks are Most Informative to Performance Prediction?

We assess how representative a model is and how informative a benchmark is, by measuring the RMSE improvements of PMF when we add the full results of a model or dataset. The most informative models and tasks are shown in Fig. 6. As observed, strong models like GPT-4, Gemini, and InternLM are more representative than weaker models. This is likely because their performance tends to deviate from the average and, being more general, they reliably reflect the difficulty level of various datasets. Interestingly, the text-to-image generation task is particularly informative. In this task, models must select the correct generated image from four candidates, and we observe that strong models, such as GPT-4, perform significantly better than others. This performance gap leads to larger errors in PMF, so including this dataset can significantly improve the PMF model.

## 6. Conclusion and Future Work

Our framework estimates unknown LVLM performances across tasks using PMF, prioritizes evaluations based on uncertainty, and introduces some enhancements to address the sparse data issue. Our study could result in significant savings in development time and computation costs. We highlight two limitations. First, modeling different LVLM evaluation settings, such as 5-shot and chain-of-thought could extend our framework. While some preliminary study is shown in Section 5.2, more exploration is left as future work. Second, it is challenging to apply our framework to new models or datasets without any performance data. We discuss this in the supplementary material (Section C.5).

## Acknowledgments

We would like to extend our deepest appreciation to Qiyuan Zhang, Caixia Zhou, Weijian Deng, Xingjian Leng, Evan Markou, Yunzhong Hou, Yicong Hong, Taojun Lin, Jiahao Zhang, Sam Bahrami, Shu Zou, Zeyu Zhang, Yang Yang, Jiahao Ma, and all our other lab colleagues for their invaluable support throughout this project. Their collaborative efforts, insightful discussions, and constructive feedback have been crucial in shaping and improving our paper.

This work was supported by an Australian Research Council (ARC) Linkage grant (project number LP210200931). Qinyu Zhao was supported by OpenAI Researcher Access Program (0000005453) and Google Cloud Research Credits Program (353970026).

## Impact Statement

This paper presents work whose goal is to advance the field of Machine Learning. There are many potential societal consequences of our work, none which we feel must be specifically highlighted here.

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

## A. Estimation of Evaluation Cost

To estimate evaluation cost of LVLMs, we evaluate a representative model, Qwen2.5-VL-Instruct 7B (Wang et al., 2024b) based on LMMs-Eval (Zhang et al., 2024b) with one A100 GPU. The results are shown in Table 5.

As shown in the first and second columns, although vllm significantly accelerate the evaluation, the computational cost is still high. Assuming that we are comparing 5 7B models on 10 similar-scale benchmarks, the evaluation will take around 21.7 hours. Besides, in our experiments, vllm usually leads to slight performance decrease.

LMMs-Lite (a coreset method) significantly reduces evaluation cost, indicating reducing evaluation cost is still valuable in practice. As shown in Section 5.3, coreset methods may get inaccurate results, while our method improves them.

## B. Comprehensive Evaluation of LVLMs

We provide a comprehensive overview of the datasets and LVLMs used in our study. Detailed dataset information can be found in Table 6 and 7, while the model profiles are presented in Tables 8 and 9.

A heatmap illustrating the model ranking across datasets is shown in Fig. 7. Additionally, the correlation analysis of performance scores is illustrated in Fig. 8 and 9. Notably, even within the same model family, such as the LLaVA series, the rankings between models do not exhibit a strong correlation. Datasets tend to have much more consistent ranking correlations, suggesting that models performing well on one dataset are likely to rank highly on others.

Besides, we investigate the impact of different latent dimensions in the PMF models and find that a relatively small latent dimension, around 10, is sufficient. As shown in Fig. 10, increasing the latent dimension reduces the RMSE on the training data to zero due to overfitting, but it does not lead to significant improvements in RMSE on the testing data. Additionally, when we extract the singular values of the score matrix, we observe that the top singular values are much larger than the rest, indicating that most of the information is captured by a few dimensions. This suggests a high degree of similarity in performance scores.

## C. Further Experimental Results

### C.1. Comparing PMF and Deep Learning-Based Methods

We evaluate Deep Matrix Factorization (DMF) (Arora et al., 2019) and Graph Convolutional Matrix Completion (GCMC) (Berg et al., 2017) for comparison.

For DMF, we use MSE loss and the Adam optimizer. The

learning rate is 1e-3 and the batch size is 256. The embedding dimension of each user or item is 10, which is the same for PMF. We train DMF for 200 epochs and the result of the best epoch is reported.

For GCMC, we refer to the GitHub implementation (https://github.com/riannevdberg/gc-mc). Dropout ratio is 0.7, learning rate is 0.01, hidden units are [500, 75] in 1st and 2nd layers, accumulation method is "stack", the number of basis functions is 2, and the model is trained for 200 epochs. We note two main issues when using GCMC:

First, GCMC handles discrete rating levels and treats each rating level as a separate class (see Section 2.3 in Berg et al. (2017)), which is not suitable for our setting, because we also use continuous ratings like the BART scores. To address this, we only use LVLM benchmarks with accuracy as the main metric, and discretize accuracy into 101 classes, i.e., 0, ..., 100. Second, when training data is sparse, some classes (for example, accuracy = 67) do not occur in training set, leading to running errors in the code.

Table 10 summarizes the average results from 10 experiments with different random seeds. As shown, PMF demonstrates superior performance on our dataset compared to DMF and GCMC. We notice that deep matrix completion methods are commonly applied in recommender systems, where there are usually thousands of users and items, with millions of samples. But in our setting, we have 108 models, 176 datasets, and 19K samples. Thus, we build our method on a simple but strong baseline, PMF, rather than neural networks, which are possibly more data-hungry.

### C.2. Detailed Performance Evaluations of PMF and PTF

We present detailed performance evaluations of PMF in Table 11 and PTF in Table 12. As shown, our methods consistently outperform the baselines. In scenarios where performance data is sparse, our enhancements significantly improves the prediction accuracy of PMF.

We also include the following ranking-based metrics.

**Spearman's rank correlation.**

**Kendall rank correlation.**

**Precision@K.** The proportion of the predicted top 1 model that fall within the top K positions of the ground-truth ranking. For example, our method predicts LLaVA-1.5 is the best on the task and it will be correct if LLaVA-1.5 is within top 5 of the ground-truth ranking for Precision@5.

**Recall@K.** The proportion of the ground-truth best ones that is correctly retrieved within our top K predicted results. For instance, if LLaVA-1.5 is the best model, it must be within the top 3 predictions of our method for Recall@3.

As shown in Table 13, ranking-based metrics also supports the advantage of our methods. Baseline methods typically achieve very high precision but low recall, while our methods provide more balanced precision and recall, as well as improved rank correlation.

"Unc 50%" and "Unc 30%" mean keeping the 50% or 30% most confident predictions based on our estimated uncertainty, which further improve the estimation accuracy. But note that this may lead to fewer predicted results.

### C.3. Combining Coreset Methods with Our Approach

We sample different ratios of coreset samples from the benchmarks, and the results are shown in Table 14.

### C.4. What Can We Tell Based on Vision Encoders?

Detailed results are shown in Fig. 11.

### C.5. Generalization to New Models and Datasets

We also investigate the models' ability to generalize to new models and datasets without any performance scores for training. As illustrated in Fig. 12, using model and dataset profiles provides slight improvement for new models or datasets. However, when both the model and dataset are entirely new, performance falls below the Global Mean baseline. But we argue that this situation is rare in practice. Some initial performance scores are usually available when a model or dataset is released, and the community usually reports more performance scores in subsequent works.

*Table 5.* **Evaluation Cost** in different settings.

| Setting | Basic | vllm | LMMs-Lite | Larger model | Video benchmarks | Test-time scaling |
|---|---|---|---|---|---|---|
| Use vllm | N | Y | Y | N (no memory) | Y | Y |
| Model Size | 7B | 7B | 7B | 32B | 7B | 7B |
| Dataset | MMBench En | MMBench En | MMBench En Lite | MMBench En | VideoMME | MathVision |
| Num of Samples | 6.72K | 6.72K | 500 | 6.72K | 2.7K | 5.14K |
| Real Performance? | Y | Possibly lower | Estimated | Y | Y | Y |
| Time | 24min | 13min | 1min | 45min | >2h* | 1h |
| Time per sample | 0.21s | 0.11s | 0.11s | 0.40s | >1s | 0.70s |

* Video loading and preprocessing are the bottleneck.

*Table 6.* **Dataset Information.** Our study utilizes 36 benchmarks. For larger benchmarks such as SEED-2, we divide them into sub-datasets based on task categories. To reduce computational costs, we subsample certain benchmarks.

| Benchmark | No. of Datasets | No. of Samples for GPT and Gemini | No. of Samples for Other Models | Download URL |
|---|---|---|---|---|
| SEED 2 (Li et al., 2023a) | 27 | 2606 | 24371 | https://huggingface.co/datasets/lmms-lab/SEED-Bench-2 |
| MME (Fu et al., 2023) | 14 | 1000 | 2374 | https://huggingface.co/datasets/lmms-lab/MME |
| MMBench CN (Liu et al., 2023b) | 20 | 1994 | 4329 | https://huggingface.co/datasets/lmms-lab/MMBench |
| MMBench EN (Liu et al., 2023b) | 20 | 1994 | 4329 | https://huggingface.co/datasets/lmms-lab/MMBench |
| MMMU (Yue et al., 2024) | 30 | 900 | 900 | https://huggingface.co/datasets/lmms-lab/MMMU |
| CMMMU (Zhang et al., 2024a) | 6 | 573 | 900 | https://huggingface.co/datasets/lmms-lab/CMMMU |
| ScienceQA (Lu et al., 2022) | 25 | 1467 | 2017 | https://huggingface.co/datasets/lmms-lab/ScienceQA |
| CVBench (Tong et al., 2024a) | 4 | 400 | 2638 | https://huggingface.co/datasets/nyu-visionx/CV-Bench |
| POPE (Li et al., 2023b) | 3 | 900 | 900 | https://github.com/AoiDragon/POPE |
| DECIMER (Brinkhaus et al., 2022) | 1 | 100 | 100 | https://www.kaggle.com/datasets/juliajakubowska/decimer |
| Enrico (Leiva et al., 2020) | 1 | 100 | 100 | https://userinterfaces.aalto.fi/enrico/ |
| FaceEmotion (Goodfellow et al., 2013) | 1 | 100 | 100 | https://www.kaggle.com/datasets/msambare/fer2013 |
| Flickr30k (Young et al., 2014) | 1 | 100 | 100 | https://www.kaggle.com/datasets/hsankesara/flickr-image-dataset |
| GQA (Hudson & Manning, 2019) | 1 | 100 | 100 | https://cs.stanford.edu/people/dorarad/gqa/download.html |
| HatefulMemes (Kiela et al., 2020) | 1 | 100 | 100 | https://www.kaggle.com/datasets/parthplc/facebook-hateful-meme-dataset |
| INAT (Van Horn et al., 2018) | 1 | 100 | 100 | https://ml-inat-competition-datasets.s3.amazonaws.com/2021/val.tar.gz |
| IRFL (Yosef et al., 2023) | 1 | 100 | 100 | https://huggingface.co/datasets/lampent/IRFL |
| MemeCaps (Hwang & Shwartz, 2023) | 1 | 100 | 100 | https://github.com/eujhwang/meme-cap/tree/main |
| Memotion (Sharma et al., 2020) | 1 | 100 | 100 | https://www.kaggle.com/datasets/williamscott701/memotion-dataset-7k |
| MMIMDB (Arevalo et al., 2017) | 1 | 100 | 100 | https://huggingface.co/datasets/akshayg08/mmimdb_test |
| NewYorkerCartoon (Hessel et al., 2022) | 1 | 100 | 100 | https://github.com/nextml/caption-contest-data |
| NLVR (Suhr et al., 2017) | 1 | 100 | 100 | https://github.com/lil-lab/nlvr.git |
| NLVR2 (Suhr et al., 2018) | 1 | 100 | 100 | https://github.com/lil-lab/nlvr.git |
| NoCaps (Agrawal et al., 2019) | 1 | 100 | 100 | https://huggingface.co/datasets/akshayg08/NocapsTest |
| OKVQA (Marino et al., 2019) | 1 | 100 | 100 | https://okvqa.allenai.org/download.html |
| OpenPath (Huang et al., 2023) | 1 | 100 | 100 | https://huggingface.co/datasets/akshayg08/OpenPath |
| PathVQA (He et al., 2020) | 1 | 100 | 100 | https://github.com/UCSD-AI4H/PathVQA |
| Resisc45 (Cheng et al., 2017) | 1 | 100 | 100 | https://www.kaggle.com/datasets/happyyang/nwpu-data-set |
| Screen2Words (Wang et al., 2021) | 1 | 100 | 100 | https://www.kaggle.com/datasets/onurgunes1993/rico-dataset |
| Slake (Liu et al., 2021) | 1 | 100 | 100 | https://huggingface.co/datasets/BoKelvin/SLAKE/ |
| UCMerced (Yang & Newsam, 2010) | 1 | 100 | 100 | https://www.kaggle.com/code/apollo2506/land-scene-classification |
| VCR (Zellers et al., 2019) | 1 | 100 | 100 | https://visualcommonsense.com/download/ |
| VisualGenome (Krishna et al., 2017) | 1 | 100 | 100 | https://homes.cs.washington.edu/ ranjay/visualgenome/ |
| VQA (Antol et al., 2015) | 1 | 100 | 100 | https://visualqa.org/vqa_v1_download.html |
| VQARAD (Lau et al., 2018) | 1 | 100 | 100 | https://huggingface.co/datasets/flaviagiammarino/vqa-rad |
| Winoground (Thrush et al., 2022) | 1 | 100 | 100 | https://huggingface.co/datasets/facebook/winoground |
| MathVision (Wang et al., 2024a) | 1 | Not evaluated | 3040 | https://huggingface.co/datasets/MathLLMs/MathVision |
| EMMA (Hao et al., 2025) | 4 | Not evaluated | 2788 | https://huggingface.co/datasets/luckychao/EMMA |
| Video-MME (Fu et al., 2024) | 1 | Not evaluated | 2700 | https://huggingface.co/datasets/lmms-lab/Video-MME |
| LongVideoBench (Wu et al., 2024) | 1 | Not evaluated | 1337 | https://huggingface.co/datasets/longvideobench/LongVideoBench |

*Table 7.* **Dataset Metrics.** PMF models the main metric on the datasets, while PTF utilizes the main and other metrics (six in total) in modeling. BARTScore is proposed by Yuan et al. (2021), while BERTScore is introduced by Zhang et al. (2019).

| Benchmark | Main Metric | Other Metrics |
| --- | --- | --- |
| SEED 2 (Li et al., 2023a) | Accuracy | - |
| MME (Fu et al., 2023) | Accuracy | Precision, Recall, F1 |
| MMBench CN (Liu et al., 2023b) | Accuracy | - |
| MMBench EN (Liu et al., 2023b) | Accuracy | - |
| MMMU (Yue et al., 2024) | Accuracy | - |
| CMMMU (Zhang et al., 2024a) | Accuracy | - |
| ScienceQA (Lu et al., 2022) | Accuracy | - |
| CVBench (Tong et al., 2024a) | Accuracy | - |
| POPE (Li et al., 2023b) | Accuracy | Precision, Recall, F1 |
| DECIMER (Brinkhaus et al., 2022) | BARTScore | BERTScore |
| Enrico (Leiva et al., 2020) | BARTScore | BERTScore |
| FaceEmotion (Goodfellow et al., 2013) | BARTScore | BERTScore |
| Flickr30k (Young et al., 2014) | BARTScore | BERTScore |
| GQA (Hudson & Manning, 2019) | BARTScore | BERTScore |
| HatefulMemes (Kiela et al., 2020) | BARTScore | BERTScore |
| INAT (Van Horn et al., 2018) | BARTScore | BERTScore |
| IRFL (Yosef et al., 2023) | BARTScore | BERTScore |
| MemeCaps (Hwang & Shwartz, 2023) | BARTScore | BERTScore |
| Memotion (Sharma et al., 2020) | BARTScore | BERTScore |
| MMIMDB (Arevalo et al., 2017) | BARTScore | BERTScore |
| NewYorkerCartoon (Hessel et al., 2022) | BARTScore | BERTScore |
| NLVR (Suhr et al., 2017) | BARTScore | BERTScore |
| NLVR2 (Suhr et al., 2018) | BARTScore | BERTScore |
| NoCaps (Agrawal et al., 2019) | BARTScore | BERTScore |
| OKVQA (Marino et al., 2019) | BARTScore | BERTScore |
| OpenPath (Huang et al., 2023) | BARTScore | BERTScore |
| PathVQA (He et al., 2020) | BARTScore | BERTScore |
| Resisc45 (Cheng et al., 2017) | BARTScore | BERTScore |
| Screen2Words (Wang et al., 2021) | BARTScore | BERTScore |
| Slake (Liu et al., 2021) | BARTScore | BERTScore |
| UCMerced (Yang & Newsam, 2010) | BARTScore | BERTScore |
| VCR (Zellers et al., 2019) | BARTScore | BERTScore |
| VisualGenome (Krishna et al., 2017) | BARTScore | BERTScore |
| VQA (Antol et al., 2015) | BARTScore | BERTScore |
| VQARAD (Lau et al., 2018) | BARTScore | BERTScore |
| Winoground (Thrush et al., 2022) | BARTScore | BERTScore |
| MathVision (Wang et al., 2024a) | Accuracy | - |
| EMMA (Hao et al., 2025) | Accuracy | - |
| Video-MME (Fu et al., 2024) | Accuracy | - |
| LongVideoBench (Wu et al., 2024) | Accuracy | - |

*Table 8.* **Model Information.** Our study evaluates 108 models. For each model, we report the number of parameters in the LLM backbone, the vision encoder, and the model family that we define.

| Model | Checkpoint | No. Param. in LLM (B) | Vision Encoder | Model Family |
|---|---|---|---|---|
| BLIP2 | BLIP2-opt-2.7B | 2.7 | ViT | BLIP |
| | BLIP2-flan-t5-xxl | 11 | ViT | BLIP |
| | BLIP2-opt-6.7b-coco | 6.7 | ViT | BLIP |
| | BLIP2-opt-6.7b | 6.7 | ViT | BLIP |
| | BLIP2-flan-t5-xl | 3 | ViT | BLIP |
| InstructBLIP | InstructBLIP-Vicuna-7B | 7 | ViT | BLIP |
| | InstructBLIP-Vicuna-13B | 13 | ViT | BLIP |
| | InstructBLIP-flan-t5-xl | 3 | ViT | BLIP |
| | InstructBLIP-flan-t5-xxl | 11 | ViT | BLIP |
| MiniGPT4 | MiniGPT4-LLaMA2-7B | 7 | ViT | MiniGPT4 |
| | MiniGPT4-Vicuna0-7B | 7 | ViT | MiniGPT4 |
| | MiniGPT4-Vicuna0-13B | 13 | ViT | MiniGPT4 |
| mPLUG-Owl | mPLUG-Owl2-LLaMA2-7B | 7 | ViT | MiniGPT4 |
| | mPLUG-Owl2_1 | 7 | ViT | mPLUG-Owl |
| LLaVA | LLaVA-7B | 7 | CLIP | LLaVA |
| | LLaVA-13B | 13 | CLIP | LLaVA |
| | LLaVA-v1.6-Vicuna-7B | 7 | CLIP | LLaVA |
| | LLaVA-v1.6-Vicuna-13B | 13 | CLIP | LLaVA |
| | LLaVA-v1.6-Mistral-7B | 7 | CLIP | LLaVA |
| | LLaVA-v1.6-34B | 34 | CLIP | LLaVA |
| Cambrian-1 | Cambrian-Phi3-3B | 3 | CLIP, SigLIP, ConvNeXt, DINOv2 | Cambrian |
| | Cambrian-8B | 8 | CLIP, SigLIP, ConvNeXt, DINOv2 | Cambrian |
| | Cambrian-13B | 13 | CLIP, SigLIP, ConvNeXt, DINOv2 | Cambrian |
| | Cambrian-34B | 34 | CLIP, SigLIP, ConvNeXt, DINOv2 | Cambrian |
| Fuyu | Fuyu-8B | 8 | - | Fuyu |
| LLaMA_Adapter | LLaMA-Adapter-V2-BIAS-7B | 7 | CLIP | LLaMA-Adapter |
| | LLaMA-Adapter-V2-LORA-BIAS-7B | 7 | CLIP | LLaMA-Adapter |
| | LLaMA-Adapter-V2-LORA-BIAS-7B-v21 | 7 | CLIP | LLaMA-Adapter |
| OpenFlamingo | OpenFlamingo-3B-vitl-mpt1b | 1 | NFNet | OpenFlamingo |
| | OpenFlamingo-3B-vitl-mpt1b-langinstruct | 1 | NFNet | OpenFlamingo |
| | OpenFlamingo-4B-vitl-rpj3b | 3 | NFNet | OpenFlamingo |
| | OpenFlamingo-4B-vitl-rpj3b-langinstruct | 3 | NFNet | OpenFlamingo |
| | OpenFlamingo-9B-vitl-mpt7b | 7 | NFNet | OpenFlamingo |
| Qwen-VL | Qwen-VL-Chat | 7 | ViT | Qwen |
| | Qwen2-VL-2B-Instruct | 2 | ViT | Qwen |
| | Qwen2-VL-7B-Instruct | 7 | ViT | Qwen |
| | Qwen2.5-VL-3B-Instruct | 3 | ViT | Qwen |
| | Qwen2.5-VL-7B-Instruct | 7 | ViT | Qwen |
| | Qwen2.5-VL-32B-Instruct | 32 | ViT | Qwen |
| DeepSeek-VL | DeepSeek-VL2-tiny | 10 | CLIP | Qwen |
| | DeepSeek-VL2-small | 28 | CLIP | Qwen |
| InternLM_XComposer | InternLM-XComposer-7B | 7 | CLIP | InternLM |
| | InternLM-XComposer-vl-7B | 7 | CLIP | InternLM |
| | InternLM-XComposer2-7B | 7 | CLIP | InternLM |
| | InternLM-XComposer2-vl-1_8b | 1.8 | CLIP | InternLM |
| | InternLM-XComposer2-vl-7B | 7 | CLIP | InternLM |
| GPT4 | gpt-4o-2024-05-13 | Unknown | Unknown | GPT4 |
| | gpt-4o-2024-08-06 | Unknown | Unknown | GPT4 |
| | gpt-4o-mini-2024-07-18 | Unknown | Unknown | GPT4 |
| | gpt-4-turbo-2024-04-09 | Unknown | Unknown | GPT4 |
| Gemini | gemini-1.5-pro | Unknown | Unknown | Gemini |
| | gemini-1.5-flash | Unknown | Unknown | Gemini |

*Table 9.* **Model Information.** This is the continued table of Table 8

| Model | Checkpoint | No. Param. in LLM | Vision Encoder | Model Family |
|---|---|---|---|---|
| Prismatic | reproduction-llava-v15+7b | 7 | CLIP | prism |
| | reproduction-llava-v15+13b | 13 | CLIP | prism |
| | one-stage+7b | 7 | CLIP | prism |
| | one-stage+13b | 13 | CLIP | prism |
| | full-ft-multi-stage+7b | 7 | CLIP | prism |
| | full-ft-one-stage+7b | 7 | CLIP | prism |
| | in1k-224px+7b | 7 | ViT | prism |
| | dinov2-224px+7b | 7 | DINOv2 | prism |
| | clip-224px+7b | 7 | CLIP | prism |
| | siglip-224px+7b | 7 | SigLIP | prism |
| | clip-336px-resize-crop+7b | 7 | CLIP | prism |
| | clip-336px-resize-naive+7b | 7 | CLIP | prism |
| | siglip-384px-letterbox+7b | 7 | SigLIP | prism |
| | siglip-384px-resize-crop+7b | 7 | SigLIP | prism |
| | siglip-384px-resize-naive+7b | 7 | SigLIP | prism |
| | dinoclip-336px-letterbox+7b | 7 | CLIP, DINOv2 | prism |
| | dinoclip-336px-resize-naive+7b | 7 | CLIP, DINOv2 | prism |
| | dinosiglip-384px-letterbox+7b | 7 | SigLIP, DINOv2 | prism |
| | dinosiglip-384px-resize-naive+7b | 7 | SigLIP, DINOv2 | prism |
| | llama2+7b | 7 | CLIP | prism |
| | llama2+13b | 13 | CLIP | prism |
| | vicuna-no-cotraining+7b | 7 | CLIP | prism |
| | llama2-no-cotraining+7b | 7 | CLIP | prism |
| | train-1.25-epochs+7b | 7 | CLIP | prism |
| | train-1.5-epochs+7b | 7 | CLIP | prism |
| | train-2-epochs+7b | 7 | CLIP | prism |
| | train-3-epochs+7b | 7 | CLIP | prism |
| | llava-lvis4v+7b | 7 | CLIP | prism |
| | llava-lrv+7b | 7 | CLIP | prism |
| | llava-lvis4v-lrv+7b | 7 | CLIP | prism |
| | prism-clip-controlled+7b | 7 | CLIP | prism |
| | prism-clip-controlled+13b | 13 | CLIP | prism |
| | prism-clip+7b | 7 | CLIP | prism |
| | prism-clip+13b | 13 | CLIP | prism |
| | prism-siglip-controlled+7b | 7 | SigLIP | prism |
| | prism-siglip-controlled+13b | 13 | SigLIP | prism |
| | prism-siglip+7b | 7 | SigLIP | prism |
| | prism-siglip+13b | 13 | SigLIP | prism |
| | prism-dinosiglip-controlled+7b | 7 | SigLIP, DINOv2 | prism |
| | prism-dinosiglip-controlled+13b | 13 | SigLIP, DINOv2 | prism |
| | prism-dinosiglip+7b | 7 | SigLIP, DINOv2 | prism |
| | prism-dinosiglip+13b | 13 | SigLIP, DINOv2 | prism |
| | prism-dinosiglip-224px-controlled+7b | 7 | SigLIP, DINOv2 | prism |
| | prism-dinosiglip-224px+7b | 7 | SigLIP, DINOv2 | prism |
| | llama2-chat+13b | 13 | CLIP | prism |
| | mistral-v0.1+7b | 7 | CLIP | prism |
| | mistral-instruct-v0.1+7b | 7 | CLIP | prism |
| | phi-2+3b | 3 | CLIP | prism |
| | gemma-instruct+2b+clip | 2 | CLIP | prism |
| | gemma-instruct+2b+siglip | 2 | SigLIP | prism |
| | gemma-instruct+2b+dinosiglip | 2 | SigLIP, DINOv2 | prism |
| | gemma-instruct+8b+clip | 8 | CLIP | prism |
| | gemma-instruct+8b+siglip | 8 | SigLIP | prism |
| | gemma-instruct+8b+dinosiglip | 8 | SigLIP, DINOv2 | prism |
| | llama2-chat+7b+clip | 7 | CLIP | prism |
| | llama2-chat+7b+siglip | 7 | SigLIP | prism |
| | llama2-chat+7b+dinosiglip | 7 | SigLIP, DINOv2 | prism |
| | llama3-instruct+8b+clip | 8 | CLIP | prism |
| | llama3-instruct+8b+siglip | 8 | SigLIP | prism |
| | llama3-instruct+8b+dinosiglip | 8 | SigLIP, DINOv2 | prism |
| | mistral-instruct-v0.2+7b+clip | 7 | CLIP | prism |
| | mistral-instruct-v0.2+7b+siglip | 7 | SigLIP | prism |
| | mistral-instruct-v0.2+7b+dinosiglip | 7 | SigLIP, DINOv2 | prism |

*Table 10.* **Comparison of PMF and Deep Learning-Based Methods.**

| Method | Overall | | Acc | | BART | |
|---|---|---|---|---|---|---|
| | RMSE↓ | MAE↓ | RMSE | MAE | RMSE | MAE |
| *Test ratio: 20%* | | | | | | |
| DMF [1] | 0.225 | 0.105 | 0.086 | 0.060 | 0.538 | 0.353 |
| GCMC [2] | | | 0.187 | 0.139 | | |
| PMF | 0.193 | 0.090 | 0.074 | 0.052 | 0.461 | 0.303 |
| *Test ratio: 80%* | | | | | | |
| DMF [1] | 0.561 | 0.314 | 0.289 | 0.209 | 1.26 | 0.896 |
| GCMC [2] | | | - | - | | |
| PMF | 0.317 | 0.174 | 0.156 | 0.115 | 0.723 | 0.504 |

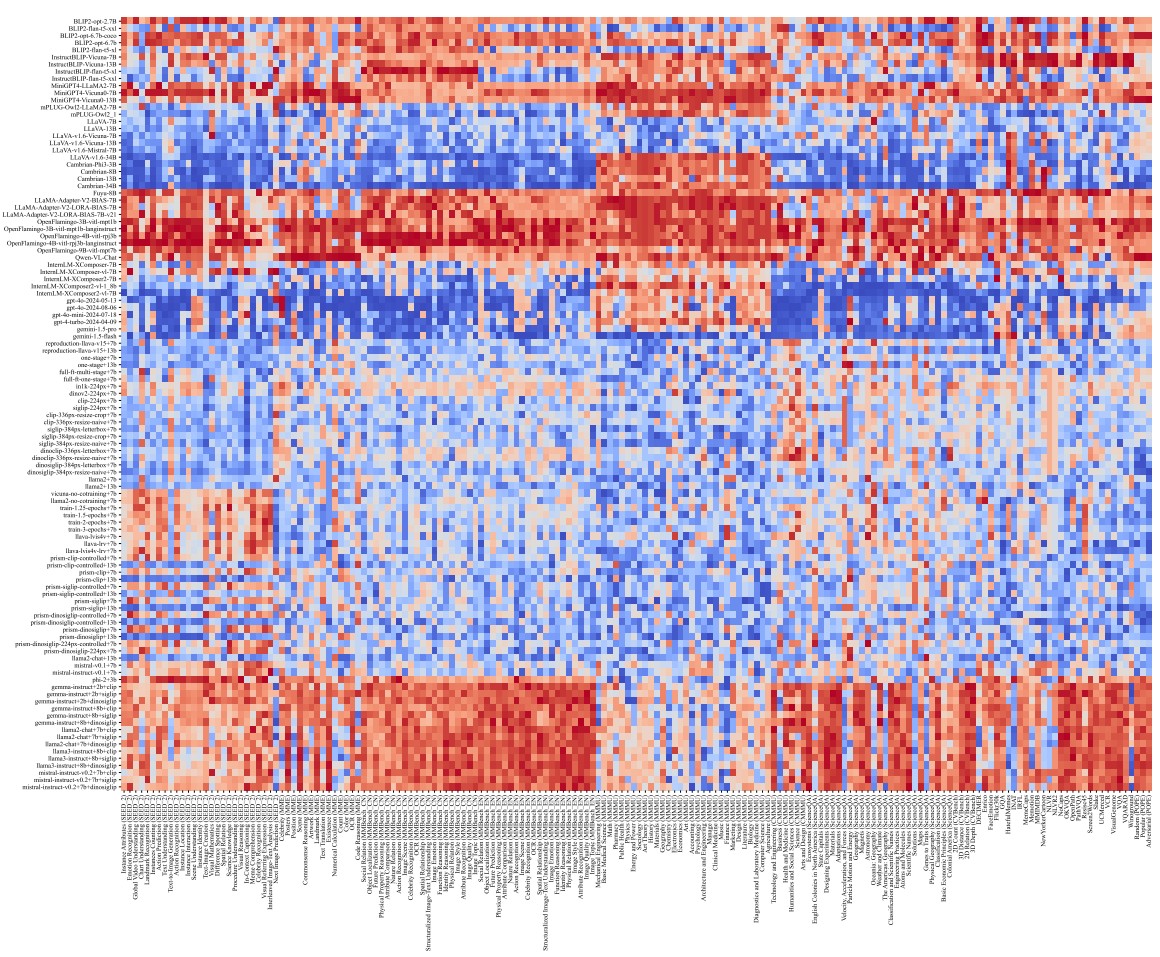

*Figure 7.* **Heatmap of Model Rankings on Each Dataset.**

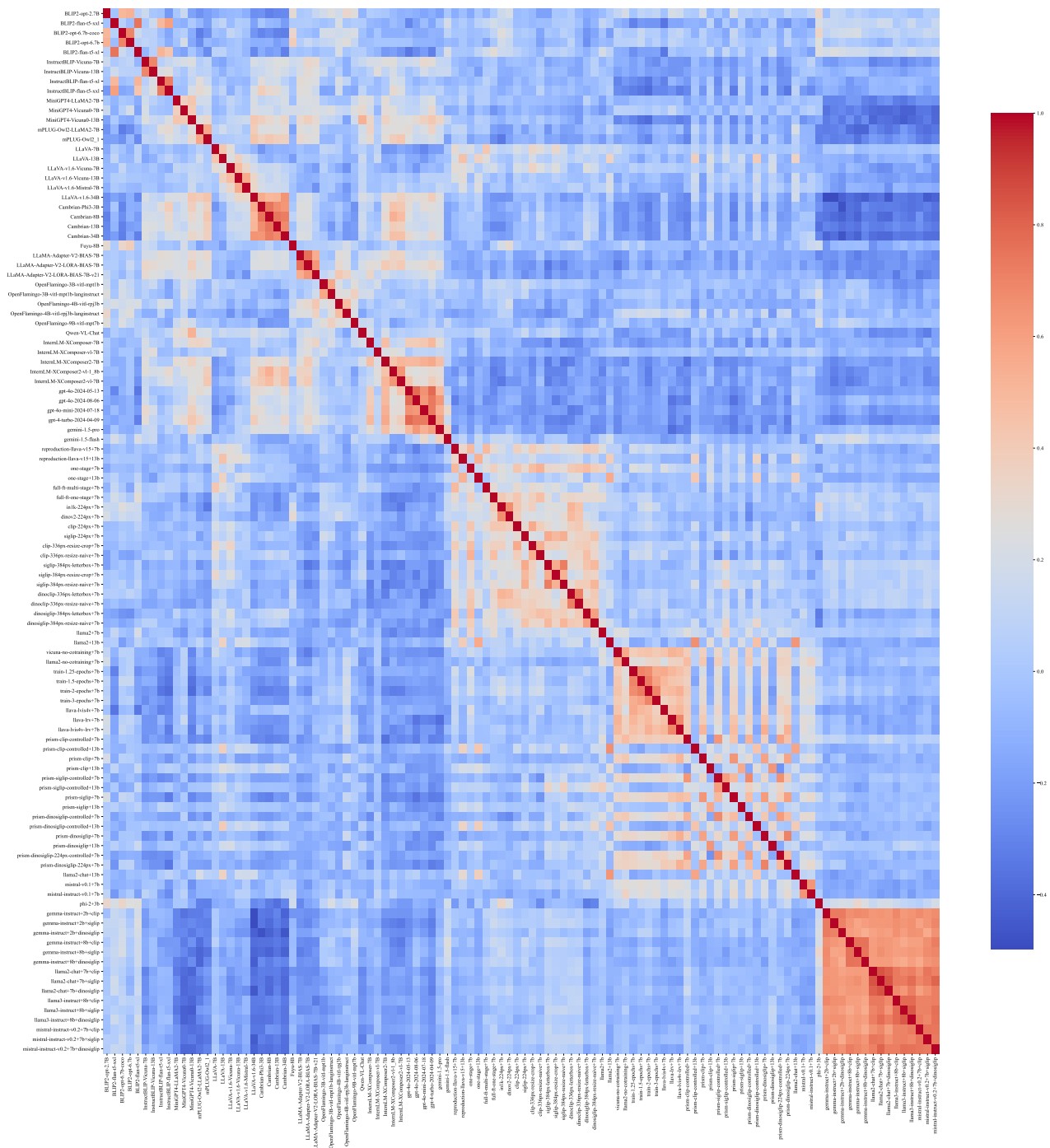

*Figure 8.* **Heatmap of Correlation in Model Ranking.**

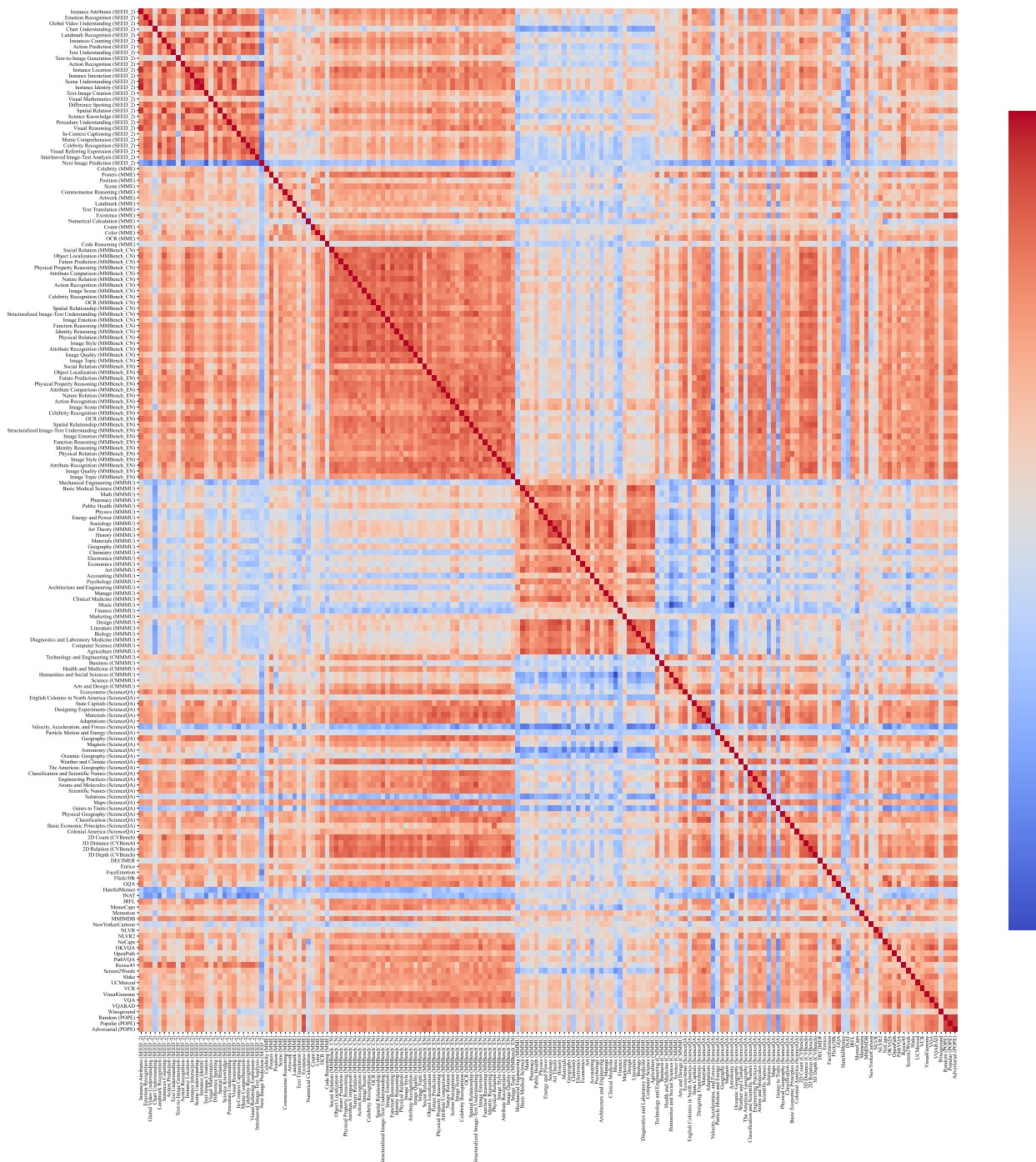

*Figure 9.* **Heatmap of Ranking Correlation on Datasets.**

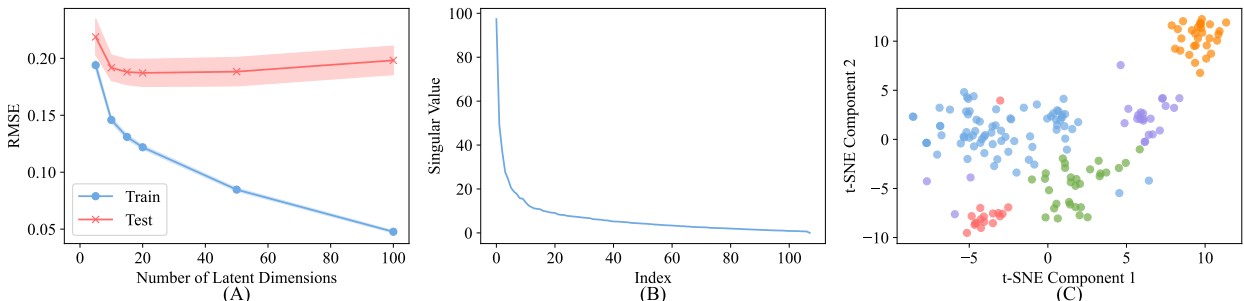

*Figure 10.* **Low-Rank Property of the Score Matrix.** (A) RMSE on the test set for PMF stabilizes when the latent dimension exceeds 15. (B) The top singular values of the performance matrix are significantly larger than the others. (C) t-SNE visualization of dataset clusters.

*Table 11.* **Detailed Performance of PMF.** Superior results are highlighted.

| Method | Overall | | Acc | | | BART | | |
|---|---|---|---|---|---|---|---|---|
| | RMSE↓ | MAE↓ | RMSE | MAE | $R^2$ ↑ | RMSE | MAE | $R^2$ |
| Test Ratio: 20% | | | | | | | | |
| Global Mean | 0.367 | 0.220 | 0.190 | 0.148 | 0.475 | 0.823 | 0.611 | 0.549 |
| Mean Of Means | 0.319 | 0.186 | 0.161 | 0.125 | 0.622 | 0.719 | 0.525 | 0.656 |
| PMF | 0.192 | 0.090 | 0.073 | 0.051 | 0.922 | 0.458 | 0.303 | 0.860 |
| Test Ratio: 40% | | | | | | | | |
| Global Mean | 0.368 | 0.220 | 0.190 | 0.149 | 0.477 | 0.829 | 0.614 | 0.551 |
| Mean Of Means | 0.320 | 0.186 | 0.161 | 0.125 | 0.623 | 0.725 | 0.527 | 0.657 |
| PMF | 0.199 | 0.095 | 0.078 | 0.056 | 0.911 | 0.474 | 0.314 | 0.853 |
| Test Ratio: 60% | | | | | | | | |
| Global Mean | 0.370 | 0.220 | 0.190 | 0.149 | 0.474 | 0.831 | 0.613 | 0.551 |
| Mean Of Means | 0.322 | 0.188 | 0.162 | 0.126 | 0.618 | 0.729 | 0.529 | 0.654 |
| PMF | 0.220 | 0.106 | 0.089 | 0.063 | 0.886 | 0.521 | 0.348 | 0.823 |
| Test Ratio: 80% | | | | | | | | |
| Global Mean | 0.373 | 0.221 | 0.193 | 0.150 | 0.462 | 0.837 | 0.612 | 0.546 |
| Mean Of Means | 0.329 | 0.191 | 0.166 | 0.128 | 0.601 | 0.742 | 0.533 | 0.643 |
| PMF | 0.258 | 0.131 | 0.114 | 0.081 | 0.812 | 0.600 | 0.407 | 0.766 |
| Test Ratio: 90% | | | | | | | | |
| Global Mean | 0.381 | 0.226 | 0.198 | 0.153 | 0.430 | 0.852 | 0.630 | 0.529 |
| Mean Of Means | 0.339 | 0.197 | 0.174 | 0.133 | 0.564 | 0.765 | 0.555 | 0.621 |
| PMF | 0.313 | 0.172 | 0.153 | 0.113 | 0.660 | 0.714 | 0.502 | 0.669 |
| Test Ratio: 95% | | | | | | | | |
| Global Mean | 0.458 | 0.278 | 0.227 | 0.182 | 0.254 | 1.041 | 0.807 | 0.297 |
| Mean Of Means | 0.385 | 0.230 | 0.191 | 0.150 | 0.474 | 0.875 | 0.672 | 0.504 |
| PMF | 0.462 | 0.276 | 0.228 | 0.180 | 0.248 | 1.052 | 0.805 | 0.282 |

*Table 12.* **Detailed Performance of PTF.** Superior results are highlighted.

| Method | Overall RMSE↓ | Overall MAE↓ | Acc RMSE | Acc MAE | Precision RMSE | Precision MAE | Recall RMSE | Recall MAE | F1 RMSE | F1 MAE | BART RMSE | BART MAE | BERT RMSE | BERT MAE |
|---|---|---|---|---|---|---|---|---|---|---|---|---|---|---|
| Test Ratio: 20% | | | | | | | | | | | | | | |
| Global Mean | 0.320 | 0.190 | 0.190 | 0.149 | 0.223 | 0.167 | 0.245 | 0.186 | 0.205 | 0.156 | 0.812 | 0.603 | 0.088 | 0.044 |
| Mean Of Means | 0.260 | 0.150 | 0.149 | 0.115 | 0.181 | 0.131 | 0.205 | 0.152 | 0.169 | 0.123 | 0.664 | 0.482 | 0.074 | 0.036 |
| PMF | 0.206 | 0.096 | 0.081 | 0.057 | 0.130 | 0.086 | 0.175 | 0.126 | 0.108 | 0.070 | 0.563 | 0.378 | 0.077 | 0.039 |
| CPTF | 0.208 | 0.099 | 0.087 | 0.060 | 0.134 | 0.089 | 0.173 | 0.123 | 0.107 | 0.070 | 0.564 | 0.379 | 0.076 | 0.039 |
| BPTF | 0.202 | 0.095 | 0.079 | 0.056 | 0.129 | 0.084 | 0.177 | 0.127 | 0.109 | 0.070 | 0.553 | 0.372 | 0.077 | 0.039 |
| BCPTF | 0.207 | 0.096 | 0.079 | 0.056 | 0.129 | 0.085 | 0.178 | 0.127 | 0.113 | 0.072 | 0.568 | 0.378 | 0.076 | 0.039 |
| Test Ratio: 40% | | | | | | | | | | | | | | |
| Global Mean | 0.323 | 0.192 | 0.190 | 0.149 | 0.223 | 0.167 | 0.247 | 0.189 | 0.213 | 0.160 | 0.818 | 0.609 | 0.091 | 0.045 |
| Mean Of Means | 0.262 | 0.151 | 0.150 | 0.116 | 0.182 | 0.132 | 0.209 | 0.156 | 0.174 | 0.126 | 0.667 | 0.486 | 0.078 | 0.038 |
| PMF | 0.210 | 0.100 | 0.083 | 0.059 | 0.132 | 0.087 | 0.181 | 0.130 | 0.112 | 0.073 | 0.572 | 0.388 | 0.081 | 0.040 |
| CPTF | 0.209 | 0.101 | 0.087 | 0.062 | 0.134 | 0.089 | 0.180 | 0.128 | 0.113 | 0.074 | 0.566 | 0.385 | 0.081 | 0.040 |
| BPTF | 0.206 | 0.100 | 0.084 | 0.060 | 0.132 | 0.086 | 0.185 | 0.133 | 0.117 | 0.077 | 0.558 | 0.379 | 0.082 | 0.041 |
| BCPTF | 0.209 | 0.100 | 0.082 | 0.059 | 0.131 | 0.086 | 0.184 | 0.131 | 0.117 | 0.076 | 0.568 | 0.384 | 0.081 | 0.040 |
| Test Ratio: 60% | | | | | | | | | | | | | | |
| Global Mean | 0.325 | 0.192 | 0.191 | 0.149 | 0.227 | 0.170 | 0.248 | 0.189 | 0.214 | 0.160 | 0.825 | 0.611 | 0.092 | 0.045 |
| Mean Of Means | 0.265 | 0.153 | 0.151 | 0.116 | 0.186 | 0.135 | 0.212 | 0.157 | 0.178 | 0.128 | 0.676 | 0.490 | 0.080 | 0.038 |
| PMF | 0.218 | 0.107 | 0.093 | 0.067 | 0.136 | 0.090 | 0.188 | 0.134 | 0.123 | 0.081 | 0.588 | 0.400 | 0.084 | 0.041 |
| CPTF | 0.219 | 0.109 | 0.098 | 0.070 | 0.141 | 0.094 | 0.187 | 0.133 | 0.125 | 0.082 | 0.588 | 0.398 | 0.083 | 0.040 |
| BPTF | 0.217 | 0.108 | 0.096 | 0.068 | 0.138 | 0.092 | 0.194 | 0.139 | 0.130 | 0.087 | 0.582 | 0.397 | 0.085 | 0.042 |
| BCPTF | 0.216 | 0.105 | 0.089 | 0.064 | 0.135 | 0.090 | 0.191 | 0.136 | 0.127 | 0.083 | 0.584 | 0.394 | 0.083 | 0.041 |
| Test Ratio: 80% | | | | | | | | | | | | | | |
| Global Mean | 0.330 | 0.194 | 0.193 | 0.150 | 0.230 | 0.171 | 0.253 | 0.191 | 0.217 | 0.162 | 0.839 | 0.619 | 0.092 | 0.046 |
| Mean Of Means | 0.277 | 0.158 | 0.155 | 0.119 | 0.198 | 0.141 | 0.225 | 0.165 | 0.186 | 0.134 | 0.709 | 0.510 | 0.084 | 0.041 |
| PMF | 0.249 | 0.128 | 0.120 | 0.087 | 0.151 | 0.103 | 0.207 | 0.148 | 0.145 | 0.098 | 0.661 | 0.457 | 0.091 | 0.044 |
| CPTF | 0.240 | 0.123 | 0.115 | 0.083 | 0.151 | 0.104 | 0.208 | 0.148 | 0.147 | 0.099 | 0.637 | 0.437 | 0.088 | 0.043 |
| BPTF | 0.239 | 0.123 | 0.116 | 0.083 | 0.152 | 0.103 | 0.212 | 0.151 | 0.151 | 0.102 | 0.630 | 0.433 | 0.090 | 0.044 |
| BCPTF | 0.236 | 0.119 | 0.108 | 0.077 | 0.147 | 0.099 | 0.208 | 0.149 | 0.147 | 0.099 | 0.627 | 0.427 | 0.089 | 0.043 |
| Test Ratio: 90% | | | | | | | | | | | | | | |
| Global Mean | 0.338 | 0.198 | 0.199 | 0.154 | 0.237 | 0.174 | 0.258 | 0.195 | 0.224 | 0.166 | 0.858 | 0.629 | 0.095 | 0.047 |
| Mean Of Means | 0.298 | 0.168 | 0.166 | 0.125 | 0.216 | 0.153 | 0.239 | 0.176 | 0.204 | 0.147 | 0.764 | 0.547 | 0.090 | 0.043 |
| PMF | 0.294 | 0.161 | 0.161 | 0.119 | 0.194 | 0.135 | 0.235 | 0.171 | 0.187 | 0.131 | 0.761 | 0.535 | 0.094 | 0.045 |
| CPTF | 0.274 | 0.147 | 0.145 | 0.105 | 0.190 | 0.133 | 0.233 | 0.168 | 0.184 | 0.128 | 0.710 | 0.492 | 0.092 | 0.043 |
| BPTF | 0.267 | 0.143 | 0.142 | 0.103 | 0.179 | 0.123 | 0.232 | 0.167 | 0.178 | 0.122 | 0.690 | 0.480 | 0.093 | 0.044 |
| BCPTF | 0.268 | 0.141 | 0.138 | 0.099 | 0.179 | 0.124 | 0.228 | 0.164 | 0.176 | 0.120 | 0.698 | 0.481 | 0.093 | 0.045 |
| Test Ratio: 95% | | | | | | | | | | | | | | |
| Global Mean | 0.404 | 0.238 | 0.228 | 0.182 | 0.251 | 0.194 | 0.270 | 0.209 | 0.240 | 0.183 | 1.058 | 0.805 | 0.101 | 0.057 |
| Mean Of Means | 0.387 | 0.217 | 0.202 | 0.158 | 0.241 | 0.182 | 0.261 | 0.198 | 0.230 | 0.172 | 1.027 | 0.772 | 0.097 | 0.056 |
| PMF | 0.403 | 0.233 | 0.223 | 0.177 | 0.244 | 0.185 | 0.269 | 0.204 | 0.236 | 0.175 | 1.059 | 0.801 | 0.101 | 0.057 |
| CPTF | 0.364 | 0.199 | 0.188 | 0.142 | 0.216 | 0.161 | 0.252 | 0.188 | 0.216 | 0.157 | 0.970 | 0.712 | 0.097 | 0.054 |
| BPTF | 0.355 | 0.196 | 0.189 | 0.144 | 0.208 | 0.153 | 0.251 | 0.188 | 0.206 | 0.148 | 0.940 | 0.687 | 0.098 | 0.055 |
| BCPTF | 0.360 | 0.196 | 0.186 | 0.141 | 0.211 | 0.156 | 0.251 | 0.186 | 0.205 | 0.148 | 0.959 | 0.702 | 0.100 | 0.056 |

*Table 13.* **Ranking-Based Metrics.** Superior results are highlighted.

| Method | Spearman↑ | Kendall↑ | Precision@1↑ | Precision@3↑ | Precision@5↑ | Recall@1↑ | Recall@3↑ | Recall@5↑ |
|---|---|---|---|---|---|---|---|---|
| Global Mean | 0.86 | 0.76 | 0.75 | 0.96 | 1.00 | 0.75 | 0.79 | 0.80 |
| Mean Of Means | 0.90 | 0.80 | 0.75 | 0.97 | 1.00 | 0.75 | 0.79 | 0.80 |
| PMF | 0.95 | 0.87 | 0.69 | 0.87 | 0.91 | 0.69 | 0.82 | 0.87 |
| BCPMF | 0.95 | 0.87 | 0.69 | 0.87 | 0.90 | 0.69 | 0.81 | 0.87 |
| BCPMF (Unc 50%) | 0.96 | 0.89 | 0.76 | 0.92 | 0.95 | 0.76 | 0.86 | 0.90 |
| BCPMF (Unc 30%) | 0.97 | 0.91 | 0.87 | 0.98 | 0.99 | 0.87 | 0.91 | 0.93 |
| Global Mean | 0.27 | 0.19 | 0.20 | 0.48 | 0.66 | 0.20 | 0.21 | 0.21 |
| Mean Of Means | 0.65 | 0.48 | 0.20 | 0.49 | 0.66 | 0.20 | 0.21 | 0.22 |
| PMF | 0.73 | 0.55 | 0.19 | 0.44 | 0.60 | 0.19 | 0.29 | 0.40 |
| BCPMF | 0.75 | 0.57 | 0.21 | 0.47 | 0.62 | 0.21 | 0.36 | 0.46 |
| BCPMF (Unc 50%) | 0.75 | 0.61 | 0.40 | 0.64 | 0.75 | 0.40 | 0.50 | 0.58 |
| BCPMF (Unc 30%) | 0.82 | 0.71 | 0.60 | 0.78 | 0.86 | 0.60 | 0.67 | 0.73 |

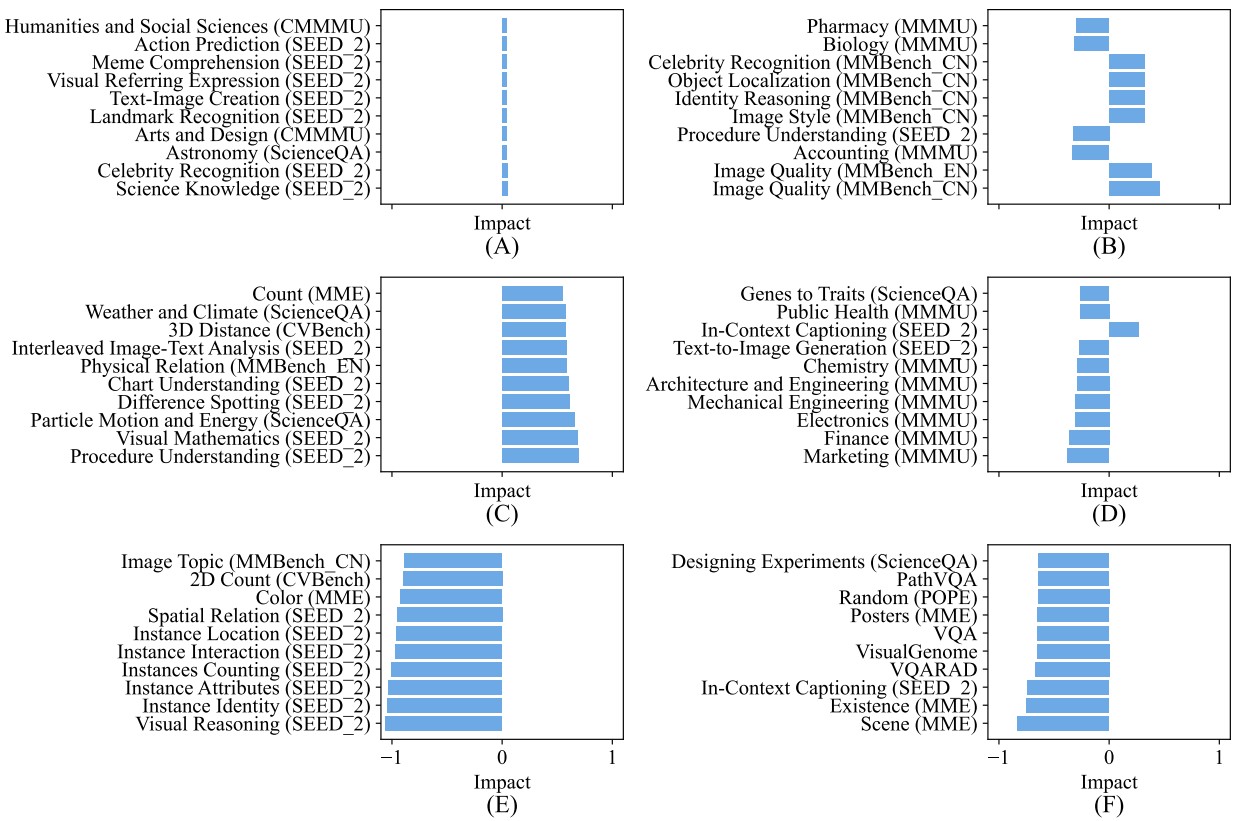

*Figure 11.* **Effect Analysis of Vision Encoders on Downstream Tasks.** We evaluate the impact of each vision encoder on downstream tasks by calculating the dot product between the feature vector of the vision encoder and the feature vector of the dataset.

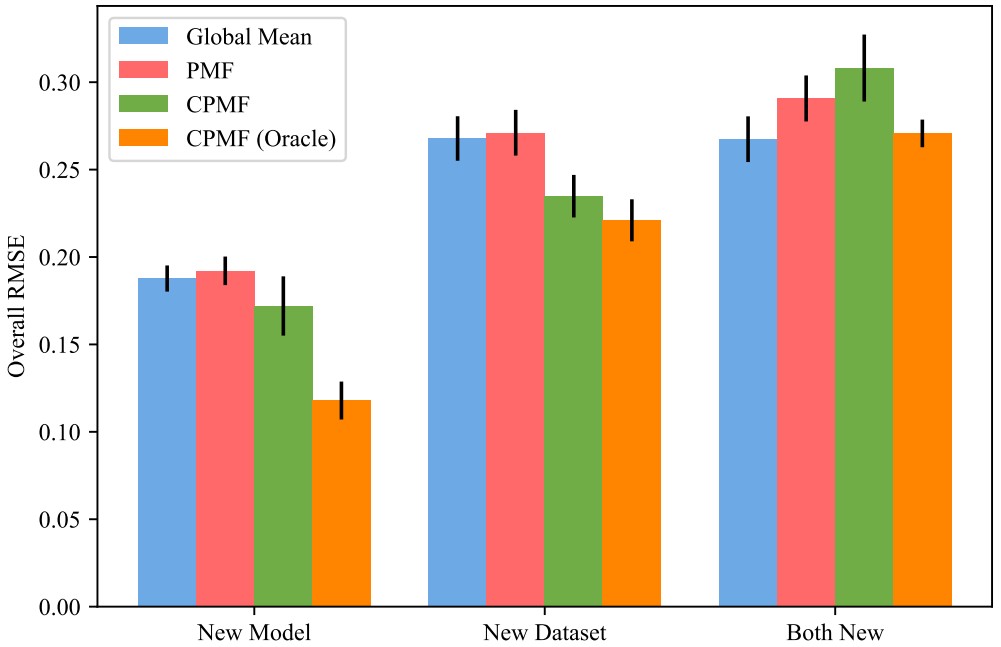

*Figure 12.* **Results on Purely New Models and Datasets.**

*Table 14.* **Combining Coreset Methods with Our Approach** with Different Sampling Ratios.

| Method | Overall | | Acc | | BART | |
|---|---|---|---|---|---|---|
| | RMSE↓ | MAE↓ | RMSE | MAE | RMSE | MAE |
| Select 5% Samples | | | | | | |
| Ours | 0.193 | 0.090 | 0.074 | 0.052 | 0.459 | 0.299 |
| Random Selection | 0.345 | 0.224 | 0.250 | 0.175 | 0.652 | 0.494 |
| + Ours (Avg) | 0.199 (-0.146) | 0.126 | 0.131 | 0.093 | 0.404 | 0.306 |
| + Ours (Unc) | 0.157 (-0.188) | 0.083 | 0.070 | 0.050 | 0.365 | 0.261 |
| Herding | 0.326 | 0.220 | 0.252 | 0.177 | 0.582 | 0.458 |
| + Ours (Avg) | 0.192 (-0.134) | 0.124 | 0.133 | 0.094 | 0.377 | 0.287 |
| + Ours (Unc) | 0.155 (-0.171) | 0.081 | 0.070 | 0.050 | 0.358 | 0.252 |
| K-Center Greedy | 0.353 | 0.231 | 0.262 | 0.182 | 0.656 | 0.498 |
| + Ours (Avg) | 0.200 (-0.153) | 0.128 | 0.137 | 0.096 | 0.394 | 0.302 |
| + Ours (Unc) | 0.154 (-0.199) | 0.082 | 0.070 | 0.050 | 0.356 | 0.258 |
| Select 10% Samples | | | | | | |
| Ours | 0.193 | 0.090 | 0.074 | 0.052 | 0.459 | 0.299 |
| Random Selection | 0.224 | 0.141 | 0.152 | 0.107 | 0.444 | 0.326 |
| + Ours (Avg) | 0.149 (-0.075) | 0.088 | 0.085 | 0.061 | 0.322 | 0.237 |
| + Ours (Unc) | 0.139 (-0.085) | 0.076 | 0.069 | 0.049 | 0.313 | 0.224 |
| Herding | 0.216 | 0.140 | 0.155 | 0.112 | 0.410 | 0.297 |
| + Ours (Avg) | 0.144 (-0.072) | 0.088 | 0.087 | 0.064 | 0.305 | 0.220 |
| + Ours (Unc) | 0.140 (-0.076) | 0.076 | 0.070 | 0.049 | 0.315 | 0.220 |
| K-Center Greedy | 0.223 | 0.142 | 0.154 | 0.109 | 0.437 | 0.322 |
| + Ours (Avg) | 0.144 (-0.079) | 0.088 | 0.086 | 0.063 | 0.306 | 0.226 |
| + Ours (Unc) | 0.138 (-0.085) | 0.077 | 0.070 | 0.049 | 0.313 | 0.224 |
| Select 15% Samples | | | | | | |
| Ours | 0.193 | 0.090 | 0.074 | 0.052 | 0.459 | 0.299 |
| Random Selection | 0.180 | 0.114 | 0.125 | 0.087 | 0.352 | 0.261 |
| + Ours (Avg) | 0.133 (-0.047) | 0.078 | 0.073 | 0.053 | 0.291 | 0.212 |
| + Ours (Unc) | 0.132 (-0.048) | 0.074 | 0.068 | 0.049 | 0.295 | 0.210 |
| Herding | 0.177 | 0.117 | 0.130 | 0.093 | 0.332 | 0.245 |
| + Ours (Avg) | 0.131 (-0.046) | 0.078 | 0.076 | 0.056 | 0.282 | 0.199 |
| + Ours (Unc) | 0.135 (-0.042) | 0.074 | 0.069 | 0.049 | 0.302 | 0.210 |
| K-Center Greedy | 0.172 | 0.111 | 0.123 | 0.087 | 0.331 | 0.239 |
| + Ours (Avg) | 0.129 (-0.043) | 0.076 | 0.073 | 0.053 | 0.281 | 0.203 |
| + Ours (Unc) | 0.131 (-0.042) | 0.074 | 0.069 | 0.049 | 0.291 | 0.208 |

