# OpenReview forum: "Can We Predict Performance of Large Models across Vision-Language Tasks?"
_ICML.cc/2025/Conference — ICML 2025 poster_

### Official Review · Reviewer_6d7s · 2025-03-06

**Overall Recommendation:** 4

**Summary:**

The paper discusses a very interesting problem, i.e., predicting the performance of MLLMs. This task is very practical because evaluating LLMs is expensive. Specifically, the authors focuses on the problem that, given a performance matrix with missing value, whether we can fill these missing ones.

## Update after rebuttal

Dear authors, I would like to express my honor to have an opportunity to review your paper and apologize for forgetting to file some areas, such as “Claims and Evidence”. Here, as the update after the rebuttal, I want to summarize my opinion and give the final recommendations.

Overall,
(1) the paper makes claims such as we can predict the performance of MLLM and these claims are well-supported by the experiments in the paper and rebuttal;
(2) after the rebuttal, I agree with that there is novelty in the proposed method, and the evaluation criterias are reasonable;
(3) I do not find any theory contributions of this paper although ICML is a conference that prefers the ML theory; but it is OK for no theoretical claims from my opinion;
(4) the experiments designs and analyses are comprehensive and detailed;
(5) the supplementary material is not required to be read by the reviewers and therefore I do not check it; and
(6) the paper mentions a strong relationshio to current literatures and all the essential references are discuessed.

In the future, I would like to recommend some extensions: (1) Exploring **why** there is a relationship between different tasks, could there be a theoratical explanations? (2) Will the proposed method can be generalizable to new released models, for example, the latest NVIDA Nemotron? (3) The degree of **needed** benchmarks, i.e., with your framework, how many benchmarks are needed in the future? If the number is limited, many new benchmarks will not be useful. (4) Including some failure cases is recommended and exploring why such cases fail. In this case, we will get clearer relationships between different MLLMs.

Finally, congratulations for such interesting work! Well done!

**Claims And Evidence:**

Yes

**Essential References Not Discussed:**

No

**Experimental Designs Or Analyses:**

Yes

**Methods And Evaluation Criteria:**

Yes

**Other Comments Or Suggestions:**

Please focus on the novelty and the relationship problem. I will raise the score to 4 if the novelty is clarified and to 5 if the relationship is justified.

**Other Strengths And Weaknesses:**

Pros:
1. I am really excited about the proposed problem. If the solutions become mature, it will save a lot of money.
2. The paper indeed tries to unveils a in-depth relationship betwwen the evaluation tasks and MLLMs.
3. As shown in Fig. 3, the proposed method is effective.


Cons:
1. The novelty. As far as my concern, the paper is lack of novelty. The paper is indeed a naive extension of Probabilistic Matrix Factorization and some related things. Please clarify the novelty in the rebuttal. I will increae the score if the novelty is clarified.
2. I do not see a strong relationship between the proposed method with the target task. Why do you focus on the evaluation of LLMs? Is it just because it is hot? It seems the proposed method is generalizable but there is no experimental results on other tasks.
3. Small mistakes: (1) the citation should be consistent: (Li & Lu, 2024). Zhang et al. (2024b) ; (2) “with limited compute” should be “with limited compution”

**Questions For Authors:**

NA

**Relation To Broader Scientific Literature:**

Yes

**Theoretical Claims:**

N/A

---

> ### Author Rebuttal · Authors · 2025-03-31
>
> > **Novelty of the paper**
>
> Thank you for your question! We would like to highlight three main contributions that reflect the novelty of our work.
>
> First, we propose and formulate the problem of LVLM performance prediction based on known performance scores. Previous works on efficient evaluation need to select a coreset within each dataset or use an unlabeled test set to estimate model performance. Instead, we explore a new direction and we show that our approach is complementary to coreset-based methods in Section 5.1.
>
> Second, we introduce model profiles and dataset profiles in Section 3.5 to enhance the prediction accuracy. For model profiles, we include features such as the number of parameters in the LLM backbone, the vision encoder type, and the LVLM family. For dataset profiles, we cluster datasets based on latent representations obtained from various models and get one-hot encoded dataset profiles. We validate three different approaches to generate these latent representations:
>
> Third, we propose an uncertainty-aware active evaluation pipeline for efficient model evaluation. As shown in Fig. 4, our uncertainty-aware evaluation consistently outperforms the random baseline under a fixed evaluation budget, especially when the budget is limited. Besides, the estimated uncertainties are correlated with the actual absolute errors, indicating the reliability of our confidence estimates.
>
> We hope that our paper can inspire future works, reduce redundancy in evaluation of large models, and contribute to more efficient development of LLMs and LVLMs.
>
> > **Relationship between the proposed method with the evaluation of LLMs. Why do you focus on the evaluation of LLMs?**
>
> Thank you for your question! We focus on LLMs and LVLMs for two main reasons. First, the evaluation of these large models is very expensive, and the number of models and datasets are growing rapidly. Second, LLMs can handle a wide variety of tasks within a single model, making it very interesting to explore the correlations of their ablities across different tasks.
>
> To reduce the evaluation costs, our paper explores a question: Can we predict unknown model performance scores based on known ones?
>
> To answer this, we draw insipration from recommender systems and adpot PMF to our question. PMF is a well-established algorithm for matrix completion. In our case, we construct a sparse performance matrix R, where each entry represents the performance score of a model on a dataset. By applying PMF, we aim to complete the performance matrix, that is, predict unknown scores. This makes PMF a natural choice for our problem.
>
> While our paper focuses on LVLMs, the proposed idea and methods are general and can be applied to other types of models and tasks as well.
>
> As an extension, we reference a related work [1], which ranks samples for efficient evaluation in image classification models. They introduce the Lifelong-ImageNet benchmark, which contains 1.98M test samples and results from 167 models.
>
> To extend our method, we validate PMF with a Sigmoid output layer to predict sample-level model accuracy (0 or 1) on Lifelong-ImageNet. Specifically, we use 10% of the performance data for training and 90% for testing. Due to the large scale of this dataset, MCMC becomes pretty slow. Thus, we use the L-BFGS-B optimization algorithm to get a maximum a posteriori estimate. The results are shown as follows.
>
> | **Method** | **Global Mean** | **Mean of Means** | **[1]*** | **PMF** |
> | -- | -- | -- | -- | -- |
> | MAE | 0.500 | 0.399 | 0.128 | 0.205 |
>
> *[1] explores a different setting to ours, so the numbers here are only for reference.
>
> More meticulous design may futher improve the performance of PMF. However, such improvements are beyond the current scope of our paper and are left for future work.
>
>
> > **Typos and mistakes**
>
> Thank you! We will carefully refine our paper and correct the typos.
>
> ---
>
> [1] Efficient Lifelong Model Evaluation in an Era of Rapid Progress. NeurIPS 2024.

---

> > ### Comment · Reviewer_6d7s · 2025-04-01
> >
> > Most of my concerns are solved, and thus I raise the score to 4.
> >
> > By the way, where is [1]?

---

> > > ### Author Response · Authors · 2025-04-01
> > >
> > > Thank you for your quick response! We truly appreciate your support and your consideration in raising the score.
> > >
> > > We put the reference at the end of our rebuttal. For your convenience, [1] is
> > >
> > > Prabhu, Ameya, et al. "Efficient Lifelong Model Evaluation in an Era of Rapid Progress." NeurIPS 2024.

---

### Official Review · Reviewer_mCcw · 2025-03-08

**Overall Recommendation:** 3

**Summary:**

This paper introduces a novel framework to predict unknown vlm  benchmark scores based on partial observation, from other LVLMs or tasks.

The problem is formulated as a matrix completion task, and the author proposes to apply probabilistic matrix factorization (PMF) with MCMC for this.

The key challenge of the vanilla baseline is the requirement of sufficient observed data, The authors augment PMF via:

- add more scores than a single accuracy (e.g., BERT score) -> PTF
- Model and dataset profiles are complied as extra information to the model.

For evaluation, the authors collect a large score matrix (108 LVLMs x 36 benchmarks), mask P% of the matrix and use the rest part to predict the masked scores. Experiment results show tthat he proposal is effective, achieving lower prediction error regarding the RMSE scores.


Further results suggest that the proposal could combine with coreset evaluation to reduce the computational costs while improving the accuracy.


## After rebuttal

The detailed response during rebuttal regarding the utility of the proposed method is somewhat convincing, and the OOD results are interesting.  Therefore, I increased my score from 2 -> 3 .

**Claims And Evidence:**

The claim that PMF is effective is well-supported by the comprehensive results such as Figure 3 and Table 1.

However, I feel the computational cost of the evaluation is somewhat exaggerated in the first paragraph, since
- We do not need to evaluate on all 50 tasks of LMMs-eval since many of them are redundant and highly correlated (as TinyBenchmark did);
- the original implementation of LMMs-eval is not very efficient (e.g., querying one sample per GPU)

I would recommend that the author report a real number with the newest LMMs-eval (with vLLM support)  on TinyBenchmark to justify the significance of this computational cost.

**Essential References Not Discussed:**

N/A

**Experimental Designs Or Analyses:**

The experiments are generally solid.

**Methods And Evaluation Criteria:**

The paper evaluates a large score matrix collected by the authors, covering diverse tasks and models.

However, as different models may require different prompts to elicit their performance, using standards templates in LMMs-eval may lead to inaccurate scores.

**Other Comments Or Suggestions:**

Sec 5.3 is interesting. Regarding the clear gain of the GPT-4 series, I am curious whether this is because many models are trained using distilled datasets from GPT-4V, such as ShareGPT-4V. The authors could separate models (trained w/ and w/o ShareGPT4V ) if see if similar conclusions could be drawn.

Typos:
Line 418: We -> we

**Other Strengths And Weaknesses:**

Pros:
- Overall, the paper is well-written and easy to follow.

Cons:

1. The actual computational cost may not be that high as stated, given the evolving software and hardware stack, rendering the problem less significant in the future;
2. From a practical perspective, we need real performance numbers to write a paper/report instead of an estimated number.
3. Most of the current benchmarks are sourced from similar original image datasets (such as COCO), and I am curious about the OOD generalization. For example, could the scores on image benchmarks using COCO images generalize to video benchmarks such as Video-MME (youtube videos)?

Points 1 and 2 are bigger ones, I am happy to discuss them with the authors.

**Questions For Authors:**

In Figure (6)B,  why do many tasks lead to the same RMSE improvements?

**Relation To Broader Scientific Literature:**

This paper proposes a new problem aiming to predict the performance scores using partially observed data points, which is novel.

**Theoretical Claims:**

N/A

---

> ### Author Rebuttal · Authors · 2025-03-31
>
> > **The computational cost**
>
> *Q: Evaluation cost with the newest LMMs-Eval implementation.*
>
> Thank you for your suggestion! We would like to clarify vllm was integrated into LMMs-Eval after our submission. We did not intend to exaggerate anything.
>
> In the rebuttal, we do not have enough time to rerun all our evaluations. Instead, we evaluate a representative model, Qwen2.5-VL-Instruct based on LMMs-Eval with 1 A100 GPU.
>
> |Setting|Basic|vllm|LMMs-Lite|Larger model|Video benchmarks|Test-time scaling|
> |-|-|-|-|-|-|-|
> |Use vllm|N|Y|Y|N (no memory) |Y|Y|
> |Model Size|7B|7B|7B|32B|7B|7B|
> |Dataset|MMBench En|MMBench En|MMBench En Lite|MMBench En|VideoMME|MathVision|
> |Num of Samples|6.72K|6.72K|500|6.72K|2.7K|5.14K|
> |Real Performance?|Y|Possibly lower|Estimated|Y|Y|Y|
> |Time|24min|13min|1min|45min|>2h*|1h|
> |Time per sample|0.21s|0.11s|0.11s|0.40s|1.3s|0.70s|
>
> **Video loading and preprocessing are the botteneck.*
>
> As shown in the first columns, although vllm significantly accelerate the evaluation, the computational cost is still high. Assuming that we are comparing 10 7B models on 10 similar-scale benchmarks, the evaluation will take around 21.7 hours. Besides, in our experiments, vllm usually leads to slight performance decrease.
>
> LMMs-Lite (a coreset method) significantly reduces the evaluation cost, indicating reducing evaluation cost is still valuable in practice. As shown in Section 5.1, coreset methods may get inaccurate results, while our method improves them.
>
> *Q. The cost will be less significant in the future*
>
> We agree that, with the development of software and hardware, evaluating old models on the old benchmarks is becoming cheaper. However, evaluation remains costly for the two main reasons.
>
> First, the growing number of models and benchmarks significantly increases the overall evaluation burden. After our submission, just from January to March, there are already several new models (e.g., the DeepSeek series, Qwen2.5 variants, MM-Eureka) and benchmarks (e.g., MathGlance, Mobile-MMLU, MathFlow, MMDT, MapBench).
>
> Second, the trend toward larger models, video LLMs, and test-time scaling techniques further raise evaluation costs, as illustrated in our table.
>
> > **We need real performance numbers to write a paper/report instead of an estimated number.**
>
> Thank you! We totally agree. Not only our paper, previous studies on predicting model performances, e.g., TinyBenchmarks and model auto-evaluation, also suffer from this problem.
>
> However, our method is still valuable during model development. For instance, when exploring the optimal video LLM design [1], the researchers evaluate their models on a reduced set of benchmarks to guide design decisions. Only at the final stage, they evaluate the model on the full benchmarks to report real performance numbers. In this case, the reseachers could use our method to reduce the evaluation cost in model development.
>
> > **OOD generalization of the method**
>
> Thank you! The question is also raised by Reviewers Yhor and dJrE. We add new models and new datasets into our pool.
>
> **New models.** Qwen2-VL-Instruct (2B, 7B), Qwen2.5-VL-Instruct (3B, 7B, 32B), DeepSeek-VL (tiny, small)
>
> **New datasets.** MathVision, EMMA, Video-MME, LongVideoBench
>
> The averaged RMSEs are shown as follows.
>
> *If we only know 20% performance of new models and new datasets*
> |Method|New Model|New Dataset|Both New|
> |-|-|-|-|
> |Global Mean|0.390|0.043|0.106|
> |Mean of Means|0.303|0.037|0.084|
> |PMF|0.326|0.032|0.047|
> |BCPMF|0.297|0.033|0.039|
>
> *If we know 50% performance of new models and new datasets*
>
> |Method|New Model|New Dataset|Both New|
> |-|-|-|-|
> |Global Mean|0.389|0.045|0.090|
> |Mean of Means|0.311|0.039|0.073|
> |PMF|0.265|0.031|0.034|
> |BCPMF|0.228|0.030|0.034|
>
> Our method shows better generalization compared to the baselines.
>
> > **Different models may require different optimal prompts**
>
> In the supplementary material (Section B.5.), we apply different evaluation settings to LLaVA and explore two ways to extend our methods. The results show that our framework can predict model performance under different evaluation settings.
>
> > **The clear gain of the GPT-4 series**
>
> Thank you! We separate models to two groups: (1) Train w/ GPT, such as LLaVA and Cambrain. (2) Train w/o GPT, such as BLIP-2 and InstructBLIP. We run 10 experiments to get the averaged results. For "Train w GPT", adding evaluation results of GPT-4o decreases 0.029 (2.48%) RMSE of PMF, while for "Train w/o GPT", it decreases 0.022 (2.04%) RMSE.  Thus, distilling knowledge from GPT may lead to similar strengths and weaknesses, which needs further exploration.
>
> > **Typos**
>
> Thank you!
>
> > **Explanation to Figure (6)B**
>
> In Fig. 6(B), the tasks in tail do not have the same RMSE improvements, but their values are relatively close. We observe that, these tasks show smaller performance gaps across models, so adding them brings limited RMSE improvement.
>
> ---
>
> [1] Apollo: An Exploration of Video Understanding in Large Multimodal Models. ArXiv.

---

> > ### Comment · Reviewer_mCcw · 2025-04-02
> >
> > Thank you for your detailed response.
> >
> > - The computational cost using vLLM is still less convincing, given that the throughput could be optimized by tuning its parameters.
> > Nevertheless, the additional inference cost is informative
> >
> > - The points that the evaluation cost would still be heavy and during the model development, the estimated numbers could be useful, are generally valid to me.
> >
> > - The OOD results with new models and datasets are exciting, validating the effectiveness of the proposal.
> >
> > As most of my concerns are addressed, I will increase my score accordingly.

---

> > > ### Author Response · Authors · 2025-04-02
> > >
> > > Thank you for your constructive comments and for increasing your score. We appreciate your recognition of the evaluation cost analysis, the practical value of our method, and the OOD results on new models and datasets.

---

### Official Review · Reviewer_dJrE · 2025-03-09

**Overall Recommendation:** 3

**Summary:**

This paper formulates the problem of predicting Large Vision-Language Model (LVLM) performance on unseen benchmarks as a sparse matrix completion task. The authors propose using Probabilistic Matrix Factorization (PMF) to predict model performance across datasets that haven't been evaluated yet. The paper introduces three key contributions: (1) a PMF-based approach to predict LVLM performance, (2) an active evaluation strategy to efficiently select which model-dataset combinations to evaluate first, and (3) an extension of PMF to Probabilistic Tensor Factorization (PTF) to handle multiple evaluation metrics simultaneously.

**Claims And Evidence:**

The paper's main claims about predicting LVLM performance using matrix factorization techniques are generally supported by empirical evidence. However, the reported MAE values (exceeding 5% in many cases) raise questions about the practical utility of these predictions for real-world evaluation scenarios. While the authors demonstrate that their approach outperforms baselines, the absolute performance may not be sufficient for confident decision-making in model selection or evaluation planning.

**Essential References Not Discussed:**

It is recommended to include more latest LVLM benchmarks in the study, to demonstrate the prediction capability of the proposed approach.

**Experimental Designs Or Analyses:**

- The experimental setup correctly partitions observed and unobserved model-dataset pairs to simulate sparse evaluations.
- The effectiveness of active evaluation is demonstrated, but further clarification is required on the robustness of the method across diverse benchmark sets.
- One potential concern is the generalizability to newer datasets. The majority of benchmarks studied are relatively early releases in 2024 (CMMMU in January, CVBench in June), while more recent ones are not tested.

**Methods And Evaluation Criteria:**

The paper primarily uses RMSE/MAE as evaluation metrics, but it may be more insightful to analyze whether the predicted model ranking is consistent with actual rankings.
Similarly, we are uncertain about how low the RMSE/MAE metrics need to be in order to provide sufficient guidance value for the model evaluation process.

**Other Comments Or Suggestions:**

- More empirical justification for metric selection (why RMSE/MAE instead of ranking-based metrics?).
- Evaluating the method on newer benchmarks (MathVision, EMMA) would provide stronger validation.
- Discussion on the implications of predictive benchmark redundancy would strengthen the paper’s impact.

**Other Strengths And Weaknesses:**

Strengths:
- The paper proposes an efficient strategy for reducing evaluation costs, which is particularly useful given the rapid development of LVLMs. The use of active evaluation is a practical addition that aligns with real-world model selection needs.

Weaknesses:
- The generalizability of predictions across benchmarks is not fully addressed—if accurate predictions indicate redundancy, does this mean newer benchmarks aren't offering meaningful novel insights? Additionally, the high MAE values suggest that the approach might not yet be sufficiently reliable for direct deployment in performance prediction tasks.

**Questions For Authors:**

1. **Metric Validity**: Have you considered ranking-based evaluation metrics instead of RMSE/MAE? Would ranking consistency be more informative for practitioners?
2. **Generalizability to New Benchmarks**: How well does your approach work on latest benchmarks, such as MathVision and EMMA?
3. **Benchmark Redundancy**: If a benchmark’s performance can be accurately predicted from previous evaluations, does this signal that the benchmark itself is redundant? How should LVLM developers interpret such cases?
4. **Prediction Accuracy**: Given the relatively high MAE values (often exceeding 5%), how confident should users be in relying on these predictions for model selection?

**Relation To Broader Scientific Literature:**

The paper builds upon Probabilistic Matrix Factorization, which is a well-established technique in collaborative filtering.
It also connects with Bayesian approaches commonly used in uncertainty estimation and low-data learning scenarios.
The work is related to efficient language model benchmarking studies such as tinyBenchmarks and Lifelong Benchmarks,
which emphasize cost-effective model evaluations.

**Theoretical Claims:**

N/A

---

> ### Author Rebuttal · Authors · 2025-03-31
>
> > **Metric Validity**
>
> Thank you for your suggestion! We include the following ranking-based metrics.
>
> **Spearman’s rank correlation.**
>
> **Kendall rank correlation.**
>
> **Precision@K.** The proportion of the predicted top 1 model that fall within the top K positions of the ground-truth ranking. For example, our method predicts LLaVA-1.5 is the best on the task and it will be correct if LLaVA-1.5 is within top 5 of the ground-truth ranking for Precision@5.
>
> **Recall@K.** The proportion of the ground-truth best ones that is correctly retrieved within our top K predicted results. For instance, if LLaVA-1.5 is the best model, it must be within the top 3 predictions of our method for Recall@3.
>
> *If we know 80% performance data for training*
>
> |Method|Spearman|Kendall|P@1|P@3|P@5|R@1|R@3|R@5|
> |-|-|-|-|-|-|-|-|-|
> |Global Mean|0.86|0.76|0.75|0.96|1.00|0.75|0.79|0.80|
> |Mean Of Means|0.90|0.80|0.75|0.97|1.00|0.75|0.79|0.80|
> |PMF|0.95|0.87|0.69|0.87|0.91|0.69|0.82|0.87|
> |BCPMF|0.95|0.87|0.69|0.87|0.90|0.69|0.81|0.87|
> |BCPMF (Unc 50%)|0.96|0.89|0.76|0.92|0.95|0.76|0.86|0.90|
> |BCPMF (Unc 30%)|0.97|0.91|0.87|0.98|0.99|0.87|0.91|0.93|
>
> Baseline methods typically achieve very high precision but low recall, while our methods provide more balanced precision and recall, as well as improved rank correlation.
>
> "Unc 50%" and "Unc 30%" mean keeping the 50% or 30% most confident predictions based on our estimated uncertainty, which further improve the estimation accuracy. But note that this may lead to fewer predicted results.
>
> *If we know 20% performance data for training*
>
> | Method|Spearman|Kendall|P@1|P@3|P@5|R@1|R@3|R@5 |
> | --|--|--|--|--|--|--|--|-- |
> | Global Mean|0.27|0.19|0.20|0.48|0.66|0.20|0.21|0.21 |
> | Mean Of Means|0.65|0.48|0.20|0.49|0.66|0.20|0.21|0.22 |
> | PMF|0.73|0.55|0.19|0.44|0.60|0.19|0.29|0.40 |
> | BCPMF|0.75|0.57|0.21|0.47|0.62|0.21|0.36|0.46 |
> | BCPMF (Unc 50%)|0.75|0.61|0.40|0.64|0.75|0.40|0.50|0.58 |
> | BCPMF (Unc 30%)|0.82|0.71|0.60|0.78|0.86|0.60|0.67|0.73 |
>
> Our method still outperforms baselines.
>
> Due to the space constraints, we omit additional results. We will update our paper accordingly.
>
> > **Generalizability to New Benchmarks**
>
> Thank you! The question is also raised by Reviewers Yhor and mCcw.
>
> We add new models and new datasets.
>
> **New models.** Qwen2-VL-Instruct (2B, 7B), Qwen2.5-VL-Instruct (3B, 7B, 32B), DeepSeek-VL (tiny, small)
>
> **New datasets.** MathVision, EMMA, Video-MME, LongVideoBench
>
> The averaged RMSEs are shown as follows.
>
> *If we only know 20% performance of new models and new datasets*
> |**Method**|**New Model**|**New Dataset**|**Both New**|
> |-|-|-|-|
> |Global Mean|0.390|0.043|0.106|
> |Mean of Means|0.303|0.037|0.084|
> |PMF|0.326|0.032|0.047|
> |BCPMF|0.297|0.033|0.039|
>
> *If we know 50% performance of new models and new datasets*
>
> |**Method**|**New Model**|**New Dataset**|**Both New**|
> |-|-|-|-|
> |Global Mean|0.389|0.045|0.090|
> |Mean of Means|0.311|0.039|0.073|
> |PMF|0.265|0.031|0.034|
> |BCPMF|0.228|0.030|0.034|
>
> Our method shows better generalization compared to the baselines. We find that generalizing to new datasets is easier than to new models. This is probably because new datasets are often very challenging, leading to generally low model performance and small RMSEs. In contrast, new models often have novel designs and remarkable improvements. This makes generalization to unseen models more difficult and needs further exploration.
>
> > **How to interpret our results?**
>
> There are two possible perspectives.
>
> **Benchmarks.** There is redundency in existing benchmarks, as also reported by [1]. Multiple benchmarks may test similar skills like math. Besides, there might be some correlation across different tasks. For instance, [2] reports that an LVLM with a stronger LLM can achieve consistently better performance on different benchmarks.
>
> **Models.** The similarity in model architectures and training may also contribute to the correlation between performance. For example, if two models use the same vision encoder, e.g., CLIP, they may have similar failure cases [3]. Besides, as commented by Reviewer mCcw, many models use training data generated from GPT-4V, possibly resulting in similar strengths and weaknesses.
>
> We will update our paper to include the discussion.
>
> > **How do users rely on our predictions?**
>
> Thank you! As shown above, the model rankings from our methods have high correlation to the ground truth. Even with very sparse training data, our methods remain more reliable than guessing based on average scores, a strategy that mimicking what a human might do. Moreover, our estimated uncertainty can further support users to make decisions. Users can focus on predictions with higher uncertainty, conduct additional evaluations, and make better model selections.
>
> ---
>
> [1] Redundancy Principles for MLLMs Benchmarks. ArXiv.
>
> [2] LLaVA-NeXT: What Else Influences Visual Instruction Tuning Beyond Data?. Blog.
>
> [3] Eyes Wide Shut? Exploring the Visual Shortcomings of Multimodal LLMs. CVPR2024.

---

### Official Review · Reviewer_Yhor · 2025-03-21

**Overall Recommendation:** 4

**Summary:**

The paper proposes a new framework for predicting the performance of large vision-language models across various tasks using probabilistic matrix factorization (PMF) with Markov chain Monte Carlo (MCMC). The framework formulates performance prediction as a matrix completion task, constructs a sparse performance matrix, and predicts unknown scores based on observed ones. The authors introduce enhancements to PMF to handle sparse data, including Bayesian PMF and incorporating model and dataset profiles. Experiments demonstrate the accuracy of PMF in predicting unknown scores and the effectiveness of active evaluation based on uncertainty.

**Claims And Evidence:**

Most claims made in the paper are generally supported by clear and convincing evidence. However, why can the performance prediction of LVLM be formulated as a matrix prediction problem? What is the underlying principle or intuition behind this approach? Is it specifically applicable to LVLM, or can it be applied to other models such as LLMs? Additionally, PMF requires that the two low-dimensional latent distributions obtained from the factorization follow a Gaussian distribution. In this context, we are dealing with different LVLM models and various benchmarks. Do these two aspects satisfy the Gaussian distribution condition?

**Essential References Not Discussed:**

N/A

**Experimental Designs Or Analyses:**

Yes

**Methods And Evaluation Criteria:**

The proposed methods and evaluation criteria make sense for the problem at hand.

**Other Comments Or Suggestions:**

N/A

**Other Strengths And Weaknesses:**

Strengths:
- The active evaluation strategy based on uncertainty is innovative and practical, potentially saving substantial computational resources.
- The comprehensive experiments and detailed analysis provide strong empirical support for the proposed methods.

Weaknesses:
- The paper does not explore the impact of different evaluation settings (e.g., varying prompts or decoding strategies) in depth.

**Questions For Authors:**

- How can the framework be extended to handle different evaluation settings, such as varying prompts or decoding strategies? Would incorporating these settings as additional profiles improve prediction accuracy?
- Can the authors provide more insights into the generalization capabilities of the framework for new models and datasets? Are there any preliminary results or ideas on how to address this limitation?

**Relation To Broader Scientific Literature:**

The paper provides a practical solution to the expensive evaluation process.

**Theoretical Claims:**

No formal theoretical proofs are provided.

---

> ### Author Rebuttal · Authors · 2025-03-31
>
> > **Explanation to our method**
>
> *Q: Why to formulate it as a matrix prediction problem? What is the underlying principle or intuition?*
>
> We are inspired by recommender systems. Imagine we are recommending movies to users: there are many users and many movies, but each user only rates a few movies. We can build a matrix where each row represents a user, each column is a movie, and each element is a rating from a user to a movie. We observe some elements in the matrix and want to estimate the unknown ones for making recommendation. This suits a matrix completion task and PMF is an effective way to solve that. In our case, we can see LVLMs as “users” while benchmarks as “movies”. We observe some performance scores of LVLMs on some benchmarks, and want to predict the unobserved ones. Thus, we formulate our problem similarly and solve it with PMF.
>
> *Q: Is it specifically applicable to LVLM?*
>
> While our paper focuses on LVLM evaluation, the methods are general and can be applied to other models and tasks as well.
>
> *Q: Do LVLM models and benchmarks satisfy the Gaussian distribution condition?*
>
> The Gaussian distribution assumption primarily serves as quadratic regularization terms in the objective function [1], which can alleviate the overfitting problem. It does not require that the latent distributions must be Gaussian. To verify this, we apply Kolmogorov-Smirnov test to check if the learned latents follow a Gaussian distribution. The p-values are all below 0.05, indicating they are not actually Gaussian.
>
> > **How can the framework be extended to handle different evaluation settings**
>
> Thank you for your question! We show some preliminary experiments in the supplementary material (Section B.5.). Specifically, we explore two ways to extend our methods.
>
> **Additional profile.** The other method is to encode evaluation settings as extra profile into PMF.
>
> **Additional models.** A straightforward way is to treat a model under different evaluation settings as different models, such as "LLaVA (Chain-of-Thought)" and "LLaVA (Beam Search)".
>
> We evaluate LLaVA-v1.5-7B on the 27 tasks in SEED-2, with the following various evaluation settings.
>
> **Image input.** (1) Default: use clean images, or (2) add Gaussian noise into the images.
>
> **Prompt.** (1) Default: prompt the model to choose option directly, (2) provide no hint, or (3) use the Chain-of-Thought (CoT) prompt.
>
> **Model decoding.** (1) Default: greedy decoding, (2) sampling with temperature = 0.2, (3) sampling with temperature = 0.5, or (4) beam search with temperature = 0.2 and the number of beams = 10.
>
> We add the results under different evaluation settings into our framework and apply PMF. The estimation accuracy is shown as follows, indicating that our framework can predict model performance under different evaluation settings.
>
> *Test Ratio: 20%* (More results are in Section B.5.)
> |**Method**|**Overall**|**Default**|**Gaussian Noise**|**No Hint**|**CoT**|**Sampling (t=0.2)**|**Sampling (t=0.5)**|**Beam Search** |
> |-|-|-|-|-|-|-|-|-|
> |Global Mean|0.119|0.112|0.105|0.090|0.117|0.127|0.109|0.111|
> |Mean of Means|0.103|0.090|0.088|0.090|0.102|0.105|0.092|0.088|
> |Ours (Profiles)|0.062|0.041|0.055|0.075|0.064|0.045|0.055|0.052|
> |Ours (Models)|0.053|0.043|0.045|0.073|0.050|0.040|0.046|0.041|
>
> > **The generalization capabilities of the framework**
>
> Thank you! The question is also raised by Reviewers dJrE and mCcw. To validate the generalization ability of the framework, we add new models and new datasets into our pool.
>
> **New models.** Qwen2-VL-Instruct (2B, 7B), Qwen2.5-VL-Instruct (3B, 7B, 32B), DeepSeek-VL (tiny, small)
>
> **New datasets.** MathVision, EMMA, Video-MME, LongVideoBench
>
> **It took several days to conduct experiments, because we need to evaluate old models on new datasets and new models on old datasets. At the end, 11 of 108 old models are not tested on the new datasets, and those results will not be included in the following test.*
>
> The averaged RMSEs are shown as follows.
>
> *If we only know 20% performance of new models and new datasets*
> |**Method**|**New Model**|**New Dataset**|**Both New**|
> |-|-|-|-|
> |Global Mean|0.390|0.043|0.106|
> |Mean of Means|0.303|0.037|0.084|
> |PMF|0.326|0.032|0.047|
> |BCPMF|0.297|0.033|0.039|
>
> *If we know 50% performance of new models and new datasets*
>
> |**Method**|**New Model**|**New Dataset**|**Both New**|
> |-|-|-|-|
> |Global Mean|0.389|0.045|0.090|
> |Mean of Means|0.311|0.039|0.073|
> |PMF|0.265|0.031|0.034|
> |BCPMF|0.228|0.030|0.034|
>
> Our method shows better generalization compared to the baselines. We observe that generalizing to new datasets is relatively easier than to new models. This is probably because new datasets are often very challenging, leading to generally lower model performance and smaller RMSEs. In contrast, new models often have novel designs and remarkable improvements. This makes generalization to unseen models more difficult and needs further exploration.
>
> ---
>
> [1] Probabilistic matrix factorization. NeurIPS 2007.

---

> > ### Comment · Reviewer_Yhor · 2025-04-06
> >
> > Thank you very much for your detailed responses, which effectively address my concerns. I suggest that the authors incorporate the content discussed in the rebuttal regarding **the inspiration from the recommender system** and **generalization to new models and datasets** into your revision. I have finally decided to raise my score to Accept.

---

> > > ### Author Response · Authors · 2025-04-06
> > >
> > > Thank you very much for your thoughtful feedback and for raising your score. We are glad that your concerns have been addressed and will incorporate the points discussed in the rebuttal into our revision.

---

### Decision · Program_Chairs · 2025-05-01

**Decision:**

Accept (poster)

**Comment:**

This paper tackles the pressing challenge of efficiently evaluating the proliferation of LVLMs by formulating performance prediction as a matrix completion task using probabilistic matrix factorization. The introduction of model and dataset profiles, together with an uncertainty‑aware active evaluation pipeline, yields consistently strong predictive accuracy across 108 models and 36 benchmarks, and generalizes well to new models and datasets. The authors have thoroughly addressed reviewer concerns—validating statistical assumptions, incorporating ranking metrics, and providing detailed cost analyses. The method promises substantial computational savings while maintaining reliability, making it a valuable contribution to the field. I recommend acceptance.